

# Mechanical compaction mechanisms in the input sediments of the Sumatra Subduction Complex- insights from microstructural analysis of cores from IODP Expedition- 362

Sivaji Lahiri[1]*, Kitty L. Milliken[2], Peter Vrolijk[3], Guillaume Desbois[1], Janos L. Urai[1]

[1]Institute of Tectonics and Geodynamics, RWTH Aachen University, Germany, Lochnerstrasse 4–20, 52056, Aachen, Germany

[2]Bureau of Economic Geology, The University of Texas at Austin, Austin, TX, 10611, USA

[3]Applied Ocean Science and Engineering, Woods Hole Oceanographic Institution, Woods Hole, MA, United States

* Corresponding author: sivaji.lahiri2@gmail.com

## Abstract

The input sediments of the North Sumatra subduction zone margin, drilled during IODP Expedition 362, exhibit remarkable uniformity in composition and grain size over the entire thickness of the rapidly deposited Nicobar Fan succession (sea-floor to 1500 mbsf depth), providing a unique opportunity to study the micromechanisms of compaction. Samples were prepared from dried core samples from sites (U1480 and U1481) by both Ar-ion cross-section polishing and broad-ion beam cutting, and imaged with a field-emission SEM. Shallowest samples (sea-floor to 28mbsf) display a sharp reduction in porosity from 80% to 52% due to collapse of large clay-domain/matrix pores associated with rotation and realignment of clay-platelets parallel to the bedding plane. The deeper succession (28mbsf to 1500mbsf) exhibits less rapid reduction in porosity from 52% to 30% by the progressive collapse of silt-adjacent larger pores by bending and subsequent sliding/fracturing of clay particles. In addition, there is a correlated loss of porosity in the pores too small to be resolved by SEM.

Clastic particles show no evidence of deformation or fracturing with increasing compaction. In the phyllosilicates, there is no evidence for pressure solution or recrystallization: thus, compaction proceeds by micromechanical processes. Increase in effective stress up to 18 MPa (~1500mbsf) causes the development of a weakly aligned phyllosilicate fabric defined by illite clay particles and mica grains, while the roundness of interparticle pores decreases as the pores become more elongated. We propose that bending of the phyllosilicates by intracrystalline slip may be the rate-controlling mechanism.

Pore size distributions show that all pores within the compactional force chain deform, irrespective of size, with increasing compactional strain. This arises because the force chain driving pore collapse is localized primarily



within the volumetrically dominant and weaker clay-rich domains; pores associated with packing around isolated
silt particles enter into the force chain asynchronously and do not contribute preferentially to pore loss over the
depth range studied.

## Introduction

Muds are fine-grained sediments (>50% of particles <63μm diameter) comprising platy detrital clay minerals and
equidimensional detrital grains such as quartz, feldspar, calcite, etc. (Nakano, 1967; Hesse, R., 1975; Sintubin,
1994). Understanding the mechanical, chemical, and microstructural properties of mud and mudstone is of great
interest for rock property prediction in basic earth science, in exploration, subsurface integrity studies and
geotechnical engineering (Yagiz, S., 2001; Aplin and Yagiz, 2011; Lazar et al., 2015). The chemical and physical
behavior of marine muds plays a critical role in defining the geometry of accretionary prisms, locating the
décollement for fault rupture (Vrolijk, 1990; Chester et al., 2013) and understanding subduction zone earthquakes
and tsunamis (Dean et al., 2010; Chester et al., 2013; Hüpers et al., 2017).
Marine mud is deposited with a highly porous isotropic fabric (Bowles, 1969; Bennett et al. 1981; Bennett et al.
1991); depositional porosity in mud is about twice as high as in sand (e.g., Velde, 1996, Lundegard, 1992). In
contrast, mudstones have low porosities, modal pore sizes measured in nm, and an absence of textural controls on
porosity (e.g. Aplin et al., 2006; Milliken et al., 2012; Milliken et al., 2013). The processes in this dramatic
evolution of porosity has similarities to compaction of sand to sandstone, comprising a combination of compaction
and cementation (Milliken and Day-Stirrat, 2013), although the much smaller, elongated phyllosilicate grains
increase the role of clay-bound water in the process (Karaborni et al., 1996). Whereas a refined and somewhat
predictive understanding exists for porosity evolution in sand and sandstones (e.g., Lander and Walderhaug, 1999;
Paxton et al., 2002; Lander et al., 2008; Adjukiewicz and Lander, 2010, Desbois et al., 2011), such a model is at
best preliminary for muds and mudstones (Pommer and Milliken, 2015; Milliken and Olson, 2017). It seems clear
that the composition of the grain assemblage importantly sets the stage for porosity evolution in muds (Milliken,
2014), cementation being the greatest in muds with abundant biogenic debris. In contrast to sandstones, however,
cementation is far less common globally in mudstones (Milliken, 2019), leading to the notion that mechanical
compaction may be far more important in muds. Establishing the expected compaction behavior for muds in a
setting of well-constrained mud properties is an essential contribution that our study hopes to serve.
Investigations of mud and mudstone compaction are usually based on proxy data, such as velocity or density,
rather than direct measurements of porosity (e.g., references in Mondol, 2007). Direct measurement of porosity
can be broadly classified into two categories: 1) laboratory experiments; (e.g., Mitchell, 1956; Bennett et al. 1981;
Vasseur et al. 1995; Mondol et al. 2007; Fawad et al. 2010; Emmanel and Day-Stirrat, 2012), and 2) studies on
natural samples (e.g., Meade, 1964; Ho et al., 1999; Aplin et al. 2003, 2006; Day-Stirrat et al., 2008; 2010; 2012;
Milliken et al, 2012, Milliken et al., 2013). A common shortcoming of studies on natural samples is the assumption
that the bulk porosity is a direct measure of compaction although porosity loss has contributions of both
compaction and cementation (Lundegard, 1992; Ehrenberg, 1989; Paxton et al., 2002), and this can only be
accomplished by petrographic inspection (Milliken and Curtis, 2017). Experimental studies generally avoid this
shortcoming by the use of lab-produced particle packs that undergo no chemical change during the experiment.



Studies of shallowly buried units (like the present study) are the ones most likely to avoid the complication of
cementation, especially if temperatures are low and bulk grain assemblages are siliciclastic (Milliken, 2014).
Previous studies report contrasting ideas about the mechanisms of mechanical compaction of mud. According to
some studies, burial compaction significantly increases the alignment of phyllosilicate (clay and mica) parallel to
the bedding planes (Bowles et al., 1969; Oertel and Curtis, 1972; Vasseur et al., 1995). Other studies state that
intense mechanical compaction has a limited impact on the development of phyllosilicate fabric in mud (Ho et al.,
1999; Aplin et al., 2006; Day-Stirrat et al., 2008; 2011).   These studies investigated the changes in particle
alignment and reduction in porosity (Ho et al., 1999; Aplin et al., 2006; Day-Stirrat et al., 2008; 2011) but without
imaging the evolution of pore morphology with increasing compactional strain.   This is where this study aims
contribute.
We received 55 mud samples from drill cores collected during IODP Expedition 362 west of the North Sumatra
subduction zone margin and investigated the evolution of petrographic microstructure and pore morphology as a
function of compactional strain. Apart from general implications for global mudrocks, we hope this investigation
will also contribute to studies that seek to predict rock properties in the deeper subsurface at the Sumatra
subduction front.

## Geological background and drilling

The Sumatra subduction zone extends 5000km from the Andaman-Nicobar Islands in the northwest to the Java-
Banda arc in the Southeast (Fig.1a and b) (Prawirodirdjo et al., 1997; Hippchen and Hyndman, 2008). The trench
of the Sumatra subduction zone (Fig.1a) developed on the subducting Indo-Australian Plate at a convergence rate
of 5.5 cm/yr in the north and 7.23 cm/yr in the South (Ghosal et al., 2014; Moeremans, and Singh, 2015).
On 26th December 2004, the west coast of Northern Sumatra recorded one of the largest earthquakes (Mw-9.3) in
the 21st century, generating a devastating Tsunami in the Indian Ocean (Ammon et al., 2005; Lay et al., 2005).
Understanding the mechanism(s) behind this unprecedented event was the central idea behind IODP Expedition
362. The main objective of the expedition was to collect core and log data of the incoming sedimentary succession
of the Indo-Australian oceanic plate to understand the seismogenic process related to the margin (Dugan et al.,
2017; McNeill et al., 2017). During the expedition in 2016, drilling was performed on two sites (U1480 and U1481)
located on the oceanic plate west of the North Sumatra subduction margin and east of the Ninety East Ridge
(Fig.1a, b) (Dugan et al., 2017). The drilling sites recovered a complete, 1.5 km thick sedimentary section from
late Cretaceous to Pleistocene down to the basement of basaltic crust (Dugan et al., 2017; McNeill et al., 2017).
The input sedimentary section of the Sumatra subduction zone comprises the distal part of the trench wedge,
Nicobar fan sequence, and pre-fan pelagic section on the basaltic crust at the bottom (Dugan et al., 2017; McNeill
et al., 2017). At Site U1480, the entire recovered section was categorized into six lithological entities, Units I to
VI (Fig. 1c) (McNeill et al., 2017). Unit I (0 to 26.72 mbsf) consists of unconsolidated calcareous clay, silty clay

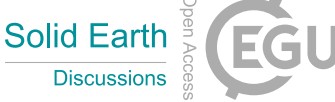

with alternating fine sand (McNeill et al., 2017). Unit II from 26.72 to 1250 mbsf consists of alternating fine-
grained sand and silty clay to silt (McNeill et al., 2017). Unit III (1250 ~ 1327 mbsf) is divided into two subunits:
Unit IIIA and IIIB (McNeill et al., 2017). Unit IIIA consist of thin to medium-bedded, gray-green or brown
mudstone and intercalated siltstone, and Unit IIIB is composed of reddish-brown tuffaceous silty claystone with
fragmented sponge spicules and radiolaria (McNeill et al., 2017). The boundary between Units IIIA and IIIB (1310
mbsf) at this site marks the base of the Nicobar Fan and the beginning of the thin pre-fan succession (Pickering et
al., 2020). Units IV, V, and VI include volcanoclastic rocks with tuffaceous sandstone, conglomerates, and basaltic
oceanic crust, respectively. For this study, we restricted ourselves to the Nicobar fan sequence that comprises Unit
I, II, and IIIA. At Site U1481, the pre-fan succession was not encountered and a Unit III, a thicker equivalent of
Subunit IIIA at Site U1480, represents the material of the lower Nicobar Fan (see Figure F15, in Site U1481 report;
McNeil et al., 2017).
X-ray diffraction (XRD) and bulk rock analysis at Site U1480 (in Units I and II) show a clay mineral assemblage
dominated by illite, with minor smectite and chlorite (Supplementary data-1) (Rosenberger et al., 2020). The
smectite content decreases with depth with the mean value of 33 wt% in Unit I and 17 wt% in Unit II (Table 1)
(Rosenberger et al., 2020). However, the relative abundance of smectite content increases sharply in Unit IIIA
with a mean value of 73 wt%. The illite percentage in the clay assemblage increases down section from Unit I to
Unit II from a mean of 49 wt% to 59 wt%; whereas decreases in Unit IIIA with a mean of 19 wt%. The
expandability of the illite/smectite mixed-layer increases down-section, which signifies an opposite trend to the
one expected for burial diagenesis (Rosenberger et al., 2020). Clay mineralogy in the lower fan muds of Unit III
of Site U1481A contains an average of 37 percent smectite and 37 percent illite (Rosenberger et al., 2020).
The Nicobar fan sequence exhibits almost compositionally homogeneous (silt/clay ratio; mostly 'silty-clay')
subunits with uniform grain size (McNeill et al., 2017), and also a history of rapid deposition (125-290 m/my;
Backman et al., 2019). The drilling sites are 255 km away from the deformation front, thus the samples are
undisturbed by tectonic faulting. In addition, the scarcity of biogenic grains and low temperatures (<68°C) make
cementation only as highly localized concretions (McNeil et al., 2017; Torres and Milliken, 2019). Such a
homogeneous sedimentary secession extending across 1.5 km depth is rare in sedimentary basins. Hence, these
samples provide us with a unique opportunity to study depth-wise variation in microstructure as a function of
vertical effective stress with few complications from multiple causes of porosity loss.

## Sampling and Methods

This study is based on two sample sets that were obtained from Sites 1480 (Holes E, F, G, and H) and 1481 (Hole
A) independently, and analyzed by slightly different methods. The first sample set (33 mud samples; depth 1.24
to 1300 mbsf) was prepared and analyzed using BIB-SEM at RWTH Aachen University, Germany. The second
sample set (22 samples; depths 6.25 to 1493.30 mbsf) was prepared using Ar-ion cross-section polishing and
imaged by field-emission SEM at the Bureau of Economic Geology (BEG) at the University of Texas at Austin.
Core description of the analysed 55 mud samples are tabulated in Supplementary data-2.





### BIB-SEM technique (analysis of the first set of samples, Aachen University)

### Sample preparation for BIB-SEM and imaging



We received 33 freeze-dried mud samples from the IODP repository, Japan (SN-1 to SN-33 in Table-1). The
samples were collected using a tube inserted perpendicular to the cut face of the drill core in such a way that the
notch of the tube identified the top of the sample so the orientation of bedding planes for each sample was known.
In Fig. 1d, a tube sample received from the IODP repository is shown, where the red line on the top of the tube
identifies the notch.  Subsamples (10 x 5 x 2 mm$^3$) were cut from the individual freeze-dried samples using a razor
blade. These subsamples were pre-polished using silicon carbide (SiC) paper to reduce the roughness of the surface
down to 10 μm. Further, Broad Ion Beam (BIB) polishing was carried out using a JEOL SM-09010 cross-section
polisher for 10 hours at 6 kV and 150 μA.  BIB reduces surface damage by removing a 100 μm thick layer to
generate a high-quality polished cross-section of 1-2 mm$^2$ with a topography less than 5 nm (Desbois et al., 2009).
After polishing, the BIB cross-sections were coated with tungsten and imaged with a Zeiss Supra 55 SEM with
SE2, BSE, and EDX detector (Supplementary data-3). SE2 images were used to image porosity, while for
identifying phases BSE images are combined with an EDX map as well as EDX point analysis. For each cross-
section, we made mosaics of hundreds of SE2 and BSE images at a magnification of 20,000x (~14.3 nm pixel
value) and 10,000x respectively, with an overlap of 20% to 30%, (Klaver et al., 2012; 2015; 2016; Hemes et al.,
2013; 2015; 2016; Laurich et al., 2014). The mosaics are stitched together using Aztec software preserving the
original pixel resolution. Finally, these stitched images are used for the segmentation of pore spaces, minerals, and
other respective analyses.

### Image segmentation and pore analysis


For quantifying porosity and pore morphology, individual SE2 image mosaics were segmented using a 'seed and
grow' algorithm (Adams and Bischof (1994)) implemented with a MatLab code (Jiang et al., 2015; Schmatz et al.,
2016) (Supplementary data-3). The 'seed and grow' algorithm works based on the difference in intensity of
greyscale value in an image (Bright = minerals, dark= pores). After automatic segmentation, individual pores in
SE2 images are manually corrected if required.
Similarly, using ImageJ software (threshold toolbox and machine learning algorithm), segmentation of the
individual mineral phases was carried out combining BSE images and EDX elemental maps. While quartz, calcite,
pyrite, mica minerals are efficiently segmented using these tools, feldspars are found difficult to segment because
of similar composition as clay (Supplementary data-4, 5 and 6). Finally, corrected pore segmented SE2 mosaics
are overlaid on the phase maps using the 'georeference' tool of QGIS (http://qgis.osgeo.org), (Supplementary data-
4, 5 and 6).

### Pore detection resolution (PPR) and representative area analysis (REA)


'Practical pore detection resolution' (PPR) indicates the pore sizes above which one can assume to detect 100%
of the pores present in the SE2 mosaic (Klaver et al., 2012). In agreement with earlier results using this instrument
(Klaver et al., 2012; 2015; 2016; Hemes et al., 2013; 2015; 2016; Laurich et al., 2014), we found PPR of ~2000



nm$^2$ and ~8500 nm$^2$ for the magnification of 20,000x and 10,000x images, respectively, corresponding to 10pixel
size in an image.
After segmenting all minerals, representative elementary area analysis (REA) was performed using the box
counting technique on mineralogical phase maps (Kameda et al., 2006; Klaver et al., 2012). Similar steps are also
followed for determining a representative elementary area for SE2 images. The estimated REA values using SE2
and BSE mosaics for the analysed 33 mudstone samples are documented in Supplementary data-7.

### Ion polishing and SEM technique (second set of samples; BEG, UT Austin).

22 Samples (SN-34 to SN-55 in Table-1) were taken shipboard from the sample half of the still-wet core in small
plastic tubes (similar to the ones used for the sample set at Aachen) inserted into the core by manual pressure. The
tubes were removed from the core and sealed in plastic bags.  In the laboratory at the BEG, sample bags were
opened and the muds were allowed to dry slowly in the tubes over several weeks. No discernible shrinkage was
observed as the dried core pieces still fully filled the tubes. The tubes were carefully removed and a small cube
(approximately 0.5 to 1 cm$^3$) was cut using a sharp knife and small hand saws; an orientation mark was placed on
the cube to indicate the bedding direction. Bed-perpendicular surfaces were prepared by Ar-ion cross-section
polishing, using the Leica EM TIC020 triple ion beam miller and coated with Ir for imaging. Manual placement
of the cut cubes into the ion mill is not precise so the ion-polished surfaces have slight variation from perpendicular
to bedding. Pore imaging was performed on the FEI Nova NanoSEM 430 using the in-lens SE detector, a 30 μm
aperture, 15 KeV accelerating current, a working distance of around 5–6 mm, and an intermediate-range sample
current (spot size = 3, mid-range for the instrument). Randomly selected views (typically 3-6) of all samples were
collected at 6kx machine magnification; additional views illustrating pore types and pore/grain relationships were
made at 10kx to 30kx (machine magnification).

### Results

### Estimating compaction strain from MAD-porosity data

Shipboard MAD (moisture and density) porosity versus depth data for mud samples exhibits a sharp reduction in
porosity from 80% to 52% from the seafloor to 28 mbsf (Fig.2a). Deeper samples display a comparatively smaller
reduction in porosity of approximately from 52% to 30% over a depth range of 28 to 1500 mbsf (Fig.2a and b).
We calculated compaction strain using the shipboard MAD porosity data following a method proposed by Nollet
et al., 2005, and subsequently used by Neagu et al., 2010 (Fig.2c and d), assuming 1D consolidation and no change
in solid volume. The compaction strain ($\varepsilon_c$) is then computed as:
$$\varepsilon_c = \frac{1-\phi_0}{1-\phi_1} \qquad\qquad\qquad \text{(Eqn-1)}$$
Here $\phi_0$ = initial porosity, and $\phi_1$ = final porosity. Our samples from sites U1480 and U1481 show no evidence of
tectonic faults (McNeill et al., 2017), supporting our assumptions. We considered the initial porosity $\phi_0$ as the
MAD porosity at 0.6 mbsf depth ($\phi_0 = 80\%$). Compaction strain following Eqn-1 (Supplementary data-7), is





plotted against depth in Fig. 2c and d. Compaction strain increases from 1 to 2.05 from the seafloor to 28 mbsf
(i.e. Unit I), and from 2.00 to 3.05 from 28mbsf to 1500 mbsf (Fig. 2c and d).
Another common measure of compaction is the intergranular volume (IGV; Paxton, 2002), which corresponds to
the sum of intergranular porosity and intergranular cement. In some mudstones, it may be necessary to calculate
IGV differently because of the presence of abundant primary intragranular pores and pore-filling bitumen
(Milliken and Olson, 2017). In our sample set, cement is absent, and IGV is taken to equal the bulk porosity from
shipboard MAD measurements.
Compactional porosity loss (COPL), referenced against the original sediment volume, is calculated from the initial
primary intergranular porosity (Pi; 80% in this case) and the IGV as follows (Lundegard, 1992; Ehrenberg, 1989):
COPL= Pi-(((100-Pi) x IGV)/(100-IGV))                                    (Eqn-2)
At an IGV of 50%, COPL is 60%; in the deepest samples in the Nicobar fan (IGV of around 30%) COPL is 70%
(supplementary data-7). Contribution of cementation (CEPL) is absent in the absence of observed cements.

### Description of grain microstructure and pore morphology

To have consistency in the data set, we prepared SE2 mosaics for all samples from the Aachen sample set at
20,000x magnification covering an average $100\times100$ $\mu m^2$ area (Supplementary data-7). In addition, to examine
the effect of magnification on BIB-SEM porosity and representative area analysis (REA), three samples (i.e. SN-
7, SN-15, and SN-29) were also imaged each at 5,000x and 10,000x magnification (Supplementary data-7),
respectively. A decrease in magnification and resolution reduces visible BIB-SEM porosity.
We observed consistent results for the REA analysis. For SE2 mosaics REA varies between $45\times45\mu m^2$ to $85\times85$
$\mu m^2$ at 20,000x magnification, and for segmented phase maps, REA varies between $90\times90$ $\mu m^2$ to $130\times130$ $\mu m^2$
at 10,000x magnification (Supplementary data-7). In the UT sample set, the standard images taken at 6kx with
machine magnification are 49.7x45.7 $\mu m^2$, so these images are also within the estimated REA range.
Six mineral phases occur in significant amounts in the Sumatra samples, as detrital particles: quartz, feldspar (K-
feldspar, albite, and Ca-plagioclase), calcite, micas (muscovite, biotite, and chlorite), and detrital clay clay-size
particles, which are dominantly illite. The average clay percentage in these mudstone samples varies between 65%
to 75%. Samples SN-1 (77%) and SN-4 (76%) are slightly enriched in clay content, whereas SN-7, SN-9, SN-17,
SN-28, SN-29, and SN-31 contain less clay (<65%) (Supplementary data-7).
Using BIB-SEM and automatic pore segmentation techniques, an average of >30,000 pores have been detected
for each individual sample in the Aachen sample set at 20,000x magnification. Correlating with the MAD data set,
the estimated BIB-SEM porosity reduces from 32% to 19% over a depth range of seafloor to 28 mbsf, while the
deeper samples display a smaller reduction from 19% to 10% over a depth range of 28 to 1500 mbsf respectively
(Fig.3a). Consistent with numerous previous studies, the results document a mismatch between bulk measured
porosity (MAD) and imaging porosity (BIB-SEM) (e.g., Hemes et al., 2013; Houben et al., 2014; Nole et al., 2016;



Olkar et al., 2019) (Supplementary data-7). We plotted BIB-SEM porosity vs MAD porosity and found an
approximately linear correlation (Fig. 3b).

## Type of pores

Intergranular pores contribute >99% of the total visible porosity (Fig.4). Intragranular pores (see below) are rare.
The size and shape of inter-granular pores change during compaction (Table 2).
Intergranular pores are classified based on grain size (irrespective of mineralogy): 1) Clay domain (matrix) pores,
and 2) Silt-adjacent pores. Based on the variation in size, clay domain pores divided further into: 1) Large clay
domain pores; pore size $>5x10^5$ nm$^2$, and the pore boundary is defined by more than three clay particles, and 2)
small clay domain pores; the pore size $<5x10^5$ nm$^2$ and generally occur in between two/three clay particles (see
further detail below). Large and small clay domain pores are classified by geometry as: 1) Elongate pores (aspect
ratio >3:1) and, 2) equant-shaped pores (aspect ratio <3:1). Elongate pores consist of: 1) linear-elongated pores,
and 2) crescent-shaped elongated pores. Examples of different clay-domain pore types are shown in Fig. 5, 6, and

250     7.

Silt-adjacent are categorized as two types: 1) large silt-adjacent pores are $>5x10^5$ nm$^2$, and pore boundaries are
defined by more than three particles; and 2) small silt-adjacent pores include pore sizes $<5x10^5$ nm$^2$, and pore
boundaries are defined by two/three particles (see further detail on the modal sizes of these pore types below).
Large and small silt-adjacent pores are either: 1) Equant shaped (aspect ratio <3:1) or 2) elongated (aspect ratio
>3:1). Further, elongated silt-adjacent pores consist of: 1) linear-shaped elongated pores and 2) crescent-shaped
elongated pores. These pore types are highlighted in Fig. 5, 6, and 7.

## Change in inter-particle pore morphology with depth

### *Seafloor to 28mbsf (Unit I)*

The shallow mud samples in Unit I are unconsolidated and highly porous (Fig. 5a; Supplementary data 8). Sample
SN-1 (1.24 mbsf) has a maximum MAD porosity of 80%. We observe three types of clay particle contacts in the
microstructure of SN-1; edge to edge (EE), edge to face (EF), and face to face (FF) contacts. Among them, EF and
FF contacts are abundant and EE contacts are rare (Table 1). The sample exhibits abundant large clay domain
pores and large silt-adjacent pores that are equant with smooth edges and a rounded pore perimeter. The sample
also contains abundant linear-elongated and equant-shaped small clay-domain pores. Crescent-shaped small clay
domain pores are rare in the microstructure of this sample. Equant-shaped, small silt-adjacent pores are abundant.
In addition, linear elongated and crescent-shaped small, silt-adjacent pores are also commonly observed (Fig.5a;
Table 2).
With increasing compaction strain ($\varepsilon_c$ = 1.119) and depth (5.1 mbsf; Supplementary data 8), porosity (MAD)
reduces to 75% and corresponding COPL=19% (sample SN-2; Fig. 5b). The microstructure of SN-2 displays
almost similar characteristics as observed in the earlier sample SN-1, although there are fewer large clay domain
pores in SN-2 than SN-1. Linear elongated and equant-shaped small clay domain pores are common (Table 2), but





crescent-shaped small clay domain pores are rare. The microstructure of SN-2 exhibits abundant equant-shaped
large and small silt-adjacent pores.
With an increase in compaction strain to $\varepsilon_c$ ~2.00 (28 mbsf), the sample microstructure is dominated by FF contacts
(Fig. 5c), and EE and EF contacts are rare (Table 3). Additionally, large clay-domain pores become sparse or
infrequent in the microstructure (Fig. 8). Crescent-shaped, small clay domain pores in the microstructure are rare,
whereas equant-shaped small clay domain pores are common. Both small and large silt-adjacent pores exhibit
equant shapes (Fig. 8d, e and f). The sample analysed at the base of Unit I (SN-6; 28 mbsf) contains rare large
clay-domain pores and abundant FF contacts (Fig. 5c; MAD porosity = 54% and COPL = 55%).
*28 mbsf to 1500 mbsf (Units II and III)*
Mud samples from the lower part of the Nicobar fan section are more compacted than shallower samples. We
analyzed a total of 29 samples using BIB-SEM at Aachen and 18 samples using the field emission SEM at UT
Austin from this section. An increase in compactional strain from 2.00 to 3.15 over a depth range of 28 to 1500
mbsf causes a porosity reduction (MAD) of 54% to 28%, and the corresponding change in average COPL is 55%
to 72%. The microstructure of these samples is dominated by FF contacts among clay particles; EF and EE contacts
are rare (Table 3; Fig. 6 and 7 b, c). All samples exhibit abundant small linear-elongated clay domain pores between
two parallel clay sheets (Fig. 8b). Equant-shaped small, clay domain pores are rarely observed below 150 mbsf
depth ($\varepsilon_c$>2.4). Crescent-shaped, small, clay domain pores are rare at shallow depth but become abundant with an
increase in compactional strain $\varepsilon_c$ >2.95 (871.87 mbsf) as the surrounding clay particles are bent (Fig.6;
Supplementary data 9). In addition, large clay domain pores in these samples are rarely observed in the vicinity of
silt clasts (Table 2).
Below 100 mbsf ($\varepsilon_c$ =2.20), silt-adjacent small pores are dominantly equant shaped, but below 300 mbsf ($\varepsilon_c$ >2.5)
silt-adjacent small pores are dominantly linear-elongated (Fig. 8e). Crescent-shaped, small, silt-adjacent pores are
common in all samples. Large silt-adjacent pores are dominantly equant above 200 mbsf depth ($\varepsilon_c$ <2.40) and
commonly linear-elongate below 400 mbsf depth ($\varepsilon_c$ >2.5) (Fig.8f). It appears that due to an increase in
compactional strain the shape of the silt-adjacent pores changes from equant to linear-elongated (Table 2). In
samples with more silt, equant-shaped small and large, silt-adjacent pores can persist at greater depths (Fig. 8e and
f).
Below 28 mbsf ($\varepsilon_c$ >2.0), the number of large silt-adjacent pores in the microstructure decreases. Comparing
samples SN-8 (74.07 mbsf and $\varepsilon_c$ =2.09) and SN-32 (1267.14mbsf and $\varepsilon_c$ =3.15) illustrates how the number of
large, silt-adjacent pores decreases with depth (Fig. 6a, and c) when the clay fraction (Supplementary Data 7) is
comparable.  This relationship is apparent even in samples separated by a smaller depth difference (SN-49 from
959.14 mbsf and SN-55 from 1433.36 mbsf; Fig. 7b and c).  While the number of large pores diminishes, the
maximum size of the large silt-adjacent pores remains constant ($10^7$ nm$^2$; Supplementary data 10).



### Intra-particle and intra-crystalline pores

Intraparticle pores are observed in microfossils, authigenic pyrite framboids, equant-shaped dolomite, calcite, quartz, mica, and lithic grains. Two different types of intraparticle pores were found in microfossils; elongated pores (Supplementary data 11a and c) and large rounded pores (Supplementary data 11b and d). The typical size of the elongated and Rounded pores in microfossils ($10^5$ nm$^2$) are larger than elongated pores ($10^3$ nm$^2$). While intraparticle pores in microfossils and most detrital grains are present at deposition, intraparticle pores in many mica grains form during burial and are considered secondary pores. Intraparticle pores in micas have cleavage-parallel, elongated pores with an aspect ratio >7 (Supplementary data 11e), and are interpreted to open by bending of the micas during compaction (Supplementary data 11f).

### Variation in the orientation of pores and grains due to compactional strain

We examined the change in orientation of the long axis of pores with increasing compaction strain. For all segmented pores, the angle between the long axis and the bedding plane was determined and plotted in rose diagrams (Supplementary data-12). Samples from the seafloor to 28 mbsf exhibit a weak preferred orientation of the long axis of pores with maxima oriented obliquely to the bedding planes. However, below 28 mbsf the samples have a preferred orientation of the long axis of pores aligned subparallel to the bedding plane Further, due to an increase in vertical effective stress down section below 28 mbsf in Units II and III, the degree of preferred alignment of the long axis of pores only increases a small amount (Supplementary data-12).

We determined the angle between the long axis of individual silt grains and the bedding plane for all samples and plotted the angle in a rose diagram (Supplementary data-12). For quartz, feldspar, and calcite the degree of preferred orientation of the long axis of grains changes little with depth. However, the rose diagrams obtained for mica show a weak maxima parallel to the bedding plane and several submaxima oriented obliquely to the bedding plane above 28 mbsf. Preferred alignment of the long axis of mica grains increases at 28 mbsf with a strong maximum oriented parallel to bedding plane. Below 28 mbsf, further increase in the degree of preferred alignment is small.

### Size distribution of pores

Pore size distributions (Fig.9) of shallow samples (Unit I) are trimodal. Sample SN-1 has peaks between $10^5$ to $10^6$ nm$^2$, $10^6$ to $10^7$ nm$^2$, and $10^7$ to $10^8$ nm$^2$, and SN-2 has peaks from $10^4$ to $10^5$ nm$^2$, $10^5$ to $10^6$ nm$^2$, and $10^6$ to $10^7$ nm$^2$. These three pore size regimes correspond to the small clay domain and silt-adjacent pores, large clay domain pores, and large silt-adjacent pores. Samples from Units II and III exhibit bimodal pore size distributions (SN-10, SN-26, and SN-33 in Fig.9). SN-10 has a peak between $10^5$ to $10^6$ nm$^2$, corresponding to small clay domain and silt-adjacent pores, and $10^6$ to $10^7$ nm$^2$, reflecting large silt-adjacent pores. Large clay domain pores are absent from samples below 28mbsf depth (Units II and III) based on the pore size distributions combined with image analysis. At the shallow depth, contribution to total porosity by larger silt adjacent pore is greater compared to the contribution by small clay domain pores (Fig.9 e and g). The contribution of large, silt-adjacent pores to total porosity diminishes with depth. Hence, at greater depth, contribution to total porosity by larger silt adjacent pore is less compared to small clay domain pores (Fig.9i).



Pore size distributions follow a power-law shown on a double logarithmic graph following the equations (Klaver
et al., 2012; 2015; 2016; Hemes et al., 2013; 2015; 2016; Laurich et al., 2014):
$\frac{N_i}{b_i S_{mosaic}} = C S_{Pore}{}^{D}$                                                   (Eqn-3)
$\log\left(\frac{N_i}{b_i S_{mosaic}}\right) = -D.\log(S_{pore}) + Log\ C$                       (Eqn-4)
Where Ni= number of pores with area Spore, bi= bin size, Smosaic= surface area of the current mosaic,
C=constant, and D= power-law exponent. The resulting power-law exponent (D) varies between 1.70 to 2.00
(Supplementary data-7).

### 348    Effect of texture on porosity, pore morphology, and orientation of pores

We analyzed six samples (SN-7, SN-9, SN-17, SN-28, SN-29 and SN-31) that are enriched in silt content compared
to the rest of the mud samples (Supplementary data-7). Silt content has a positive correlation to the total SEM
porosity. For example, sample SN-29 (1172.88 mbsf) exhibits a BIB-SEM porosity of 14% whereas other samples
from a similar depth with less silt exhibit an average BIB-SEM porosity of 12% (Supplementary data-7) at 20000x
magnification. The samples with greater silt content are also enriched in equant-shaped silt-adjacent larger pores
(Supplementary data 13a). We also estimated the orientation of the long axis of pores for these three samples and
plotted the obtained results as rose diagrams (Supplementary data 13b). The obtained results exhibit a relatively
weak preferred alignment of the long axis of pores with respect to the bedding planes (Supplementary data 11b).

## 357    Discussions

### 358    Effective stress vs porosity: A comparison with experimental study

To understand the consolidation mechanisms of the Sumatra sediment we estimated vertical effective stress
following the steps proposed by Hüpers et al., 2015. Following Tarzaghi and Peck (1948) vertical effective stress
is expressed as:
$\sigma_v{}' = \sigma_v - P_f$                                                          (Eqn-7)
Here $\sigma_v$ = total vertical stress caused by the overburden load, and $P_f$ = fluid pressure. To compute vertical effective
stress of a layered sediment, we use Eqn 8:
$\sigma_v{}' = \sum(\boldsymbol{\rho_s} - \boldsymbol{\rho_w}).\mathbf{g}.\Delta\mathbf{z}$                                         (Eqn-8)
where $\boldsymbol{\rho_s}$= bulk density of the sediment, $\boldsymbol{\rho_w}$= density of the pore water, $\Delta\mathbf{z}$= depth interval, and g= gravitational
acceleration. During IODP Expedition 362, drilling was performed 255 km away from the deformation front.
Although small offset strike-slip faults are evident at the seafloor and in seismic reflection profiles (McNeill et al.,
2017), the amount of strain attributed to these fault offsets is compatible with the idea that the maximum
horizonatal stress is comparable to the vertical stress; there is no evidence in seismic reflection data or from core





microstructures for thrust or reverse faults associated with a vertical least principal stress. On this basis, we assume

that vertical effective stress is the maximum principal stress. Bulk density of the sediment $\rho_s$ was obtained from

McNeill et al., (2017), and $\rho_w$ was considered as the density of sea-water i.e. 1025 kg/m$^3$ (Hüpers et al., 2015).

We plotted vertical effective stress vs MAD porosity of 55 mud samples (Fig. 10). Fawad et al. (2010)

experimentally studied the consolidation behavior of mud with varied proportions of silt and clay. While Sumatra

samples follow trends similar to those defined by Fawad et al. (2010), the experimental samples are more

compacted than natural Sumatra samples for the same silt content.

Clay mineralogy has a significant effect on the compaction behavior of mudstone (Mondol et al., 2007). Mondol

et al. (2007) performed compaction experiments using pure smectite and pure kaolinite clay particle packs; as they

represent two end members compared to other clay minerals (illite and chlorite) in terms of grain size and surface

area. Smectite is the most fine-grained clay and has the largest surface area; whereas kaolinite is the coarsest one

and has smaller surface area among all other clay mineral types (Meade, 1963; Mesri and Olson, 1971; Rieke and

Chilingarian, 1974). Hence, kaolinite is more compressible than smectite, and clay compaction gradually decreases

with increasing the proportion of small size clay particles in the sample (Mondol et al, 2007).

Fawad et al., (2010) used clay mixtures of 81% kaolinite, 14% mica, and 5% microcline grains, whereas Sumatra

mud samples are mainly composed of >73% of illite and lesser smectite, with only <16% undifferentiated chlorite

and kaolinite, and <7% quartz particles. Therefore, due to higher illite/smectite content, Sumatra muds appeared

to be less compacted compared to the experimental samples used by Fawad et al. (2010).

### BIB-SEM porosity vs MAD porosity

We note that BIB-SEM porosity is lower than the porosity found from the shipboard MAD data, however the two

measurements correlate along a line through the origin. (Fig.3b). The reason for this difference is that MAD

porosity measures the total amount of moisture in a much larger sample and accounts for pores much below the

PPR and also rare large pores not included in the 1 mm$^2$ BIB section. Earlier studies also documented and discussed

mismatch between MAD and BIB-SEM measurements (Hemes et al., 2013; Houben et al., 2014; Nole et al., 2016;

Olkar et al., 2019). We plotted estimated BIB-SEM porosity and MAD porosity data from earlier studies on BOOM

clay (Hemes et a., 2013; Olkar et al., 2019); Opalinus clay (Houben et al., 2014); and samples from Nankai trough

(Nole et al., 2016). The data for Boom clay and Opalinus clay follow similar trend as Sumatra samples, whereas

clay samples from Nankai trough shows different trend, and it could be attributed due to the difference in

magnification of imaging of Nankai samples.

In addition we plotted clay content vs the difference between the two porosities in Supplementary data 14a. We

performed regression analysis using the data set for the 33 muds samples analyzed at Aachen (Supplementary data

14b). First, only two variables i.e., BIB-SEM porosity vs MAD porosity (following Eqn-5); second, we considered

three variables MAD porosity, BIB-SEM porosity, and clay content (following the Eqn-6).

BIB-SEM porosity=a*MAD porosity + c                    (Eqn-5)



BIB-SEM porosity =a*MAD porosity + b*clay content + c                                                    (Eqn-6)
The coefficient of determinations (R2) for Eqn-5 and Eqn-6 are 0.8408 and 0.9262 respectively. These results
show that the ratio in porosity depends on depth and clay content.
For all samples the BIB-SEM pore size distribution follows power-law over an interval of three orders of
magnitude. We may extrapolate this below the practical pore resolution (PPR). (Klaver et al., 2012; Kuila and
Prasad, 2013; Wang et al., 2019). Extrapolating our data set down to 3nm pore diameter, it is found that BIB-SEM
porosity increases only up to 20%~25%. So, there is still an average mismatch of 15% to 20% between the MAD
porosity and extrapolated BIB-SEM porosity. The fall off from the normal trend in log-log pore size distribution
plots (Fig.9b) for the shallow depth (Unit-I) samples suggest that also large pores are uncounted in the data set.
The mud samples from Unit-I contains forams that are rare or absent in the deeper section (Supplementary data-
11a, b, c, and d), and missing pore volume can be attributed to the intact forams that may be missed due to the
small size of the BIB SEM sample. Moreover, shallow depth samples are richer in smectite compared to the deeper
samples, which can also somewhat influence to overestimate the MAD porosity.
## Micromechanical model for porosity reduction
### Sharp reduction in porosity at the shallow depth from the seafloor to 28 mbsf
High porosity (80% MAD; 32% BIB-SEM) in the shallowest sediments is attributed to large pores in the samples
created by abundant EE and EF particle contacts (Fig. 5a and 7a). These contacts are unstable and collapse to form
FF contacts, resulting in a rapid porosity decrease with 28m of burial (Table 3). This deformation is apparent from
the reduction in large clay domain pores observed over this interval (Fig. 8; Table 2). Collapse of pores surrounded
by EE and EF contacts is further recognized by the progressive alignment of clay particles into the bedding plane
and by the increase number and consistent orientation of elongated, small, clay domain pores. Each of these
observations is consistent with rotation of clay particles into the bedding plane as these large clay domain pores
collapse.
### Mechanism of porosity reduction from 28 mbsf to 1500 mbsf
Below 28 mbsf to >1500 mbsf, porosity continues to decrease from 50-32% (MAD) but at a reduced pace. SEM
observations suggest that this porosity decline results from the progressive loss of silt-adjacent pores with large
silt-adjacent pores lost before small ones (Fig. 8), although they remain present in common abundance to1200
mbsf. Small, clay domain pores are abundant throughout the section, and the large clay domain pores were lost
above 28 mbsf.
Within the population of silt-adjacent pores, the large, equant pores are most susceptible to collapse (Fig. 8). Large,
elongate pores persist in abundance, both in linear and crescent geometries. While it seems plausible that large,
equant pores collapse to form large, elongate pores, no corresponding increase in the elongate pore population is
observed. Large, elongate pores may collapse further and become small silt-adjacent pores. Microstructural
evidence supports the idea that large equant pores collapse as surrounding clay aggregates bend and shrink the size
of the remaining pore (Fig. 11a to f), and that the collapse results in an increasing aspect ratio of the pore. While



initial pore shrinkage may be accommodated solely by the bending of clay particles above the pore (Fig. 11),
further collapse may require clay particles to slide into the pore (Fig.11g-iii) or become fractured (Fig.11g-v) in
order to allow clay to invade the pore (model shown in Fig.11 is elaborately described in Supplementary data-15).
Small, silt-adjacent pores also become less abundant with burial, but the transition occurs deeper than the large
pores, and small, silt-adjacent pores remain common throughout the section (Fig. 11). Equant, small pores are lost
like the large pores, and elongate pores remain abundant within this population subset throughout. There is a loose
correspondence between the loss of small, equant pores and an increase in elongate pores, suggesting that pore
flattening is part of the pore collapse history. The pore collapse evolution outlined for large pores (Fig. 11) appears
to hold for small pores, even though observations are more challenging.
Small, clay domain pores appear to remain resilient throughout the compaction history (Fig. 8), even though some
of these pores must become lost to account for porosity loss. Small, equant pores are lost between 100-200 m, and
this loss appears to be accommodated by an increase in elongate pores (Fig. 8). Elongate crescent pores increase
in abundance around 800 mbsf, and we interpret this to reflect folding of abundant linear elongate pores as the
overall system compacts.
Large, equant pores in the clay domain are lost within the first few 10's of meters of burial. Elongate pores appear
to form at the expense of equant pores, and there may be a reduction in pore size associated with this shape change.
Most of the pores remaining after 1500 m of burial are small, elongate pores found both in clay domain and silt-
adjacent pores.
The presence of silt particles locally redistributes the force chain of load to retain silt-adjacent, large pores
undeformed (Schneider et al., 2011). The samples with greater silt content are also enriched in equant-shaped silt-
adjacent larger pores (Supplementary data 13a) in the microstructure. Hence, and as a result, they display greater
porosity compared to other samples from similar depths intervals (Supplementary data-7).
Previous laboratory studies have emphasized the importance of clay particle rotation as a dominant mechanism for
mechanical compaction in mudstone (Bennett et al., 1981; Vasseur et al., 1995; Aplin et al., 2006; Day-Stirrat et
al., 2008; 2011). While we infer rotation is an important mechanism for mechanical compaction at the shallowest
depth where unstable EE and EF particle contacts are present, clay particle bending and sliding/fracturing are
considered more important for most of the section studied.

### Mechanical compaction of marine sediment: a conceptual model

According to Emmanuel and Day-Stirrat (2020), the reduction of pores in sedimentary rocks during compaction
is size-dependent; larger pores deform much readily than smaller pores. According to their model, larger pores
rapidly decrease in size during compaction to reduce the overall porosity of the sample. However, microstructural
analysis of Sumatra samples suggests that porosity reduction is accomplished by compaction of all pore sizes.
While large, clay domain pores are lost more quickly than large, silt-adjacent pores, silt-adjacent pores are larger
than clay domain pores. Moreover, the maximum size of pores remains almost constant irrespective of increasing
vertical effective stress/depth (Supplementary data 8) with little difference observed for the maximum pore size in



samples from 98.25 mbsf and 1299.31 mbsf.  The preservation of a constant ratio between MAD and BIB-SEM
porosity measurements (Fig. 3b) suggests that porosity loss is distributed across all pore sizes.  We infer that all
pore sizes are available for compaction for every increment of applied stress but acknowledge that the rate of pore
reduction in different size classes may proceed at different rates.
We propose a new model for the reduction in porosity in which, all pores within the force chain of load take part
in the reduction in porosity during compaction irrespective of their size. At shallow depth up to 28mbsf, larger
clay-domain pores are the most susceptible to early response during an increase in compactional strain; because
of two reasons- 1) the 'domains' defined by the clay aggregate are weaker compared to the larger rigid silt grains,
and 2) due to higher relative proportion of clay-rich regions within the mud, the force chain of load dominantly
passes through the clay aggregate. The dispersed nature of the silt-size particles and the high proportion of
phyllosilicates in the mud samples indicate that soft clay particles act as the principal load-bearing framework.
Hence, larger clay domain pores are more unstable compared to silt adjacent pores in the mud microstructure.
Similarly, below 28mbsf depth in Units II and III, under an increase in vertical effective stress, both the larger silt-
adjacent pores and smaller pores in the clay matrix that come within the force chain of load collapse. Hence, the
ratio between BIB-SEM porosity vs MAD porosity remains almost constant irrespective of the depth. All larger
silt-adjacent pores do not come within the force chain of load at the same time. Hence, some of the larger silt-
adjacent pores remained undeformed to the maximum depth of 1500mbsf depth. Therefore, the maximum size of
the larger silt adjacent larger pores remains almost constant irrespective of the depth/vertical effective stress.
While our understanding of how different pore types is consistent with all available data, tracking the pore
evolution through additional size categories would elucidate the pore evolution in more detail.  Preliminary pore
size distribution data (Fig.9) indicate that 4 size bins exist in these samples.  Developing this approach requires
improved image analysis techniques to tie all the pore attributes together on a pore-by-pore basis for a huge number
of pores.

## Compaction strain accommodation and grain-scale deformation

Deformation of clay-rich sedimentary rocks involves four possible mechanisms: 1) Particulate flow; (Morgenstern
and Tchalenko, 1967; Borradaile et al., 1981); 2) Cataclasis; (Ukar and Cloos, 2019) 3) Diffusive mass transfer;
(Blenkinsop, 2000; Fossen, 2016); 4) Intercrystalline plasticity (Blenkinsop, 2000; Fossen, 2016). The intensity
and occurrence of a particular deformation mechanism in mudstone depend on several parameters, such as
effective stress, water content, cementation, temperature (Desbois et al., 2017; Den Hartog and Spiers, 2014).
All our samples show evidence of particulate flow controlled by friction between grains.  At shallow depths, illite
platelets contacted at EE and EF junctions lose these weak bonds, and particles rotate into bedding-parallel
orientation. Once FF contacts dominate, large-scale rotations are reduced and intra-crystalline slip becomes
important.  This is best evidenced in collapse of large, silt-adjacent pores where bent clay particles overlie pores
(Fig.11). In deforming granular foam material, bending was reported as the dominant deformation mechanism for
the reduction in porosity and developing preferred alignment of the long axis of pores perpendicular to the applied
stress (Elliott et al., 2002, Zhou et al., 2004; Samsudin et al., 2017; Zakaria et al., 2018) (review of these earlier



studies on the experimental deformation of granular foam is described in supplementary data-16). Friction adheres
clay particles to the edge of pores while the middle of particles drops into the pore, resulting in bending by intra-
crystalline slip. A cartoon (Fig. 12g) illustrates the compaction mechanism associated with the bending of clay
particles. With increasing compaction strain, clay particles undergo bending, and as a result, pore area reduces and
the orientation of the pores tends to align perpendicular to the applied effective stress (Fig. 12g).

## Compaction of Sumatra input section: generalized implication for rock property evolution



The overall compaction curve obtained for Sumatra muds is comparable with the experimental study by Fawad et
al., 2010 in the context of compactional range (Fig.8). The curve shows a monoexponential decrease in porosity
with an increase in vertical effective stress, which is evidence of normal consolidation (Fawad et al., 2010;
Dutilleul et al., 2020).
The larger silt-adjacent pores seen in the deepest of these samples (1500 m burial) suggest these muds retain
considerable potential for additional compaction in deeper burial. As this marine sediment progressively
approaches greater burial at closer proximity of the accretionary prism, it will undergo further change in physical
and deformational properties (Bray and Karig, 1985). Despite the substantial compactional strain, the relatively
high porosity of the deepest sample and the survival of larger and mechanically unstable silt-margin pores suggests
that compactional stabilization has not been reached because such IGVs and pore types are not generally observed
in older and lithified mudrocks.  Based on the current understanding of subduction zone deformation behavior and
mudrock properties, it seems likely that compaction will continue to dominate the pore loss in deeper burial.
The general absence of early cementation and the corresponding dominance of compaction in the total pore loss,
is consistent with observations of other siliciclastic-dominated muds (Milliken, 2014; 2019). The trends for
intergranular volume observed across the seafloor to 1500mbsf place useful constraints on the maximum cement
volumes that theoretically could be emplaced at this depth range in sediments containing a more reactive grain
assemblage. At the depths of burial attained at the deformation front, any cementation of the Sumatra input
sediments will be limited to <30% of the rock volume, or possibly much less, as compaction is expected to continue
up to the burial temperatures that initiate grain reactions and associated cementation.

## Conclusions


Pores can be classified by size and also microstructural position. Their contribution to the total porosity is
multimodal.
Samples at shallow depth (seafloor to 28 mbsf) display a sharp reduction in porosity from 80% to 52% due to the
collapse of the large clay domain/matrix pores. Deeper samples (28 mbsf to 1500 mbsf) exhibit a smaller reduction
in porosity from 50% to 32% due to collapse of silt-adjacent pores by bending and subsequently fracturing/sliding
of clay aggregate.





Large equant pores in clay (between $10^6$ and $10^7$ nm$^2$) are rare below the first few meters of burial, after the flocculated structure collapses.

The class of large pores next to silt-size grains (between $10^4$ and $10^6$ nm$^2$) remains common to >1 km burial, irrespective of the mineralogy of the silt-sized grains, but their size decreases with depth. Small, equant pores next to silt particles are abundant in the first 100 m of burial and remain common over the whole samples depth range.

Small pores in clay domains are almost all elongated, and abundant over all observed depths. Small, crescent-shaped elongate pores increase in abundance with depth as clay particles become folded by compactional processes.

The size-independence of pore loss arises because the force chain driving pore collapse is localized primarily within the volumetrically dominant and weaker clay-rich domains; larger pores around isolated silt particles enter into the force chain somewhat randomly and asynchronously and do not contribute preferentially to pore loss over the depth range studied.

An increase in effective stress up to 18MPa (~1500 mbsf) causes the development of weakly aligned phyllosilicate fabric (defined by mica and illite clay particles) in the microstructure.

Compaction processes in our samples are dominated by granular flow (rotation and frictional sliding of illite clay particles) at shallow depths. With increasing depth, compaction is additionally accommodated by bending of clay particles.

## Data availability

High resolution SE2 and BSE images of all samples are available online at:
https://figshare.com/s/cbaada517b0b1409d575

## Authors contributions

SL and KLM performed sample preparation and BIB-SEM microscopy. SL analysed the data. JLU and GD acquired funding. JLU managed the project. PV, KLM and JLU significantly contributed to interpret the data. SL wrote the first draft of the manuscript. PV, KLM and JLU contributed for the correction and improvement of the manuscript.

## Competing interests

The authors declare that they do not have any conflict of interest.



## Acknowledgments

SL and JLU thank German Research Foundation (Deutsche Forschungsgemeinschaft [DFG] grant UR 64/19-1) for providing funding to carry out the research. IODP (International Ocean Discovery Programme) sample repository, Japan is acknowledged for providing oriented mud samples for the study. KLM acknowledges the samples and data provided by the International Ocean Discovery Program (IODP). Funding for sample preparation and SEM imaging was supported by a post-expedition award (Milliken, P.I.) from the Consortium for Ocean Leadership. SL thanks Manuel Menzel, Jop Klaver, Liene Spruženiece, and Joyce Schmatz for providing valuable time to teach BIB-SEM techniques.

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



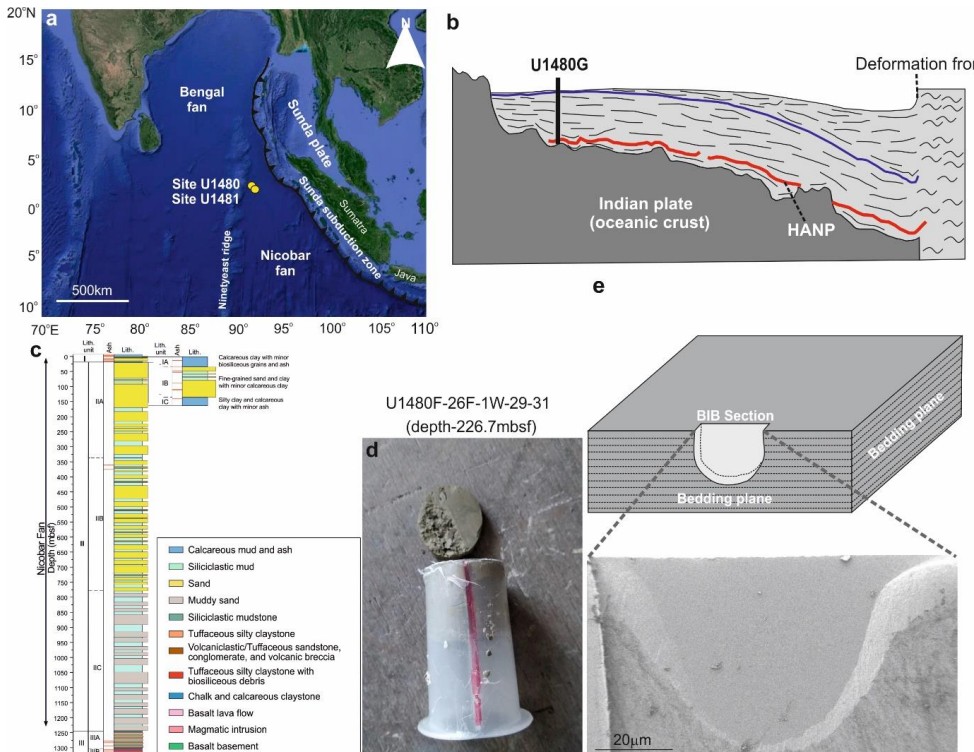


**Fig.1:** (a) Geological map of Sumatra subduction zone and location of U1480 and U1481 drilling sites (created from © Google Maps). (b) Location of drilling site U1480 in sectional view (adapted from seismic profile after Hüpers et al., 2017). (c) Lithostratigraphic units encountered at Site U1480 (adapted after McNeill et al., 2017). (d) Representative tube sample received from IODP repository, Japan. Red-colored line on tube surface represents notch used to denote orientation of samples collected from drill core. (e) Representative BIB cross-section polished perpendicular to bedding planes.










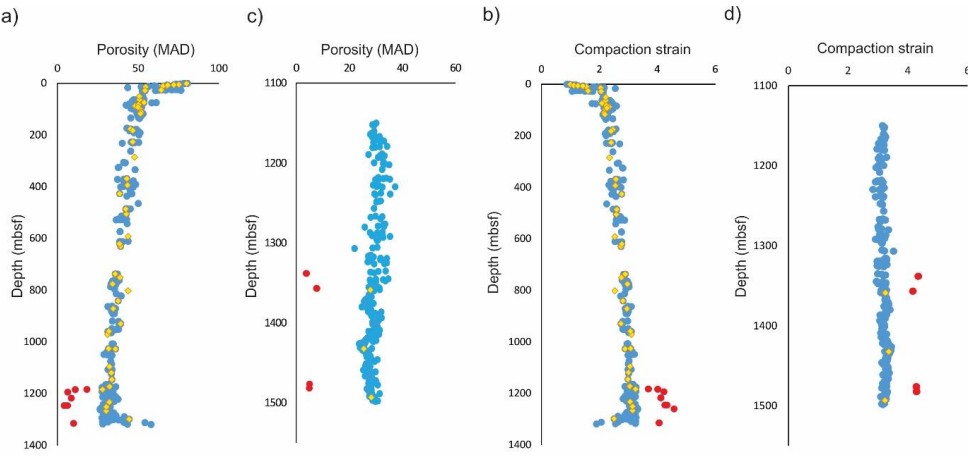

**Fig.2:** (a) and (b) Shipboard MAD (Moisture and density) porosity vs depth data for mudstone samples recovered from Sites U1480 and U1481; (c) and (d) Cross-plot diagrams for estimated compaction strain vs depth corresponding to samples recovered from Sites U1480 and U1481. Yellow-colored symbols in (a), (b), (c), and (d) show 55 mud samples analyzed at RWTH-Aachen and BEG, Austin. Red-colored points are cemented (concretion) samples.

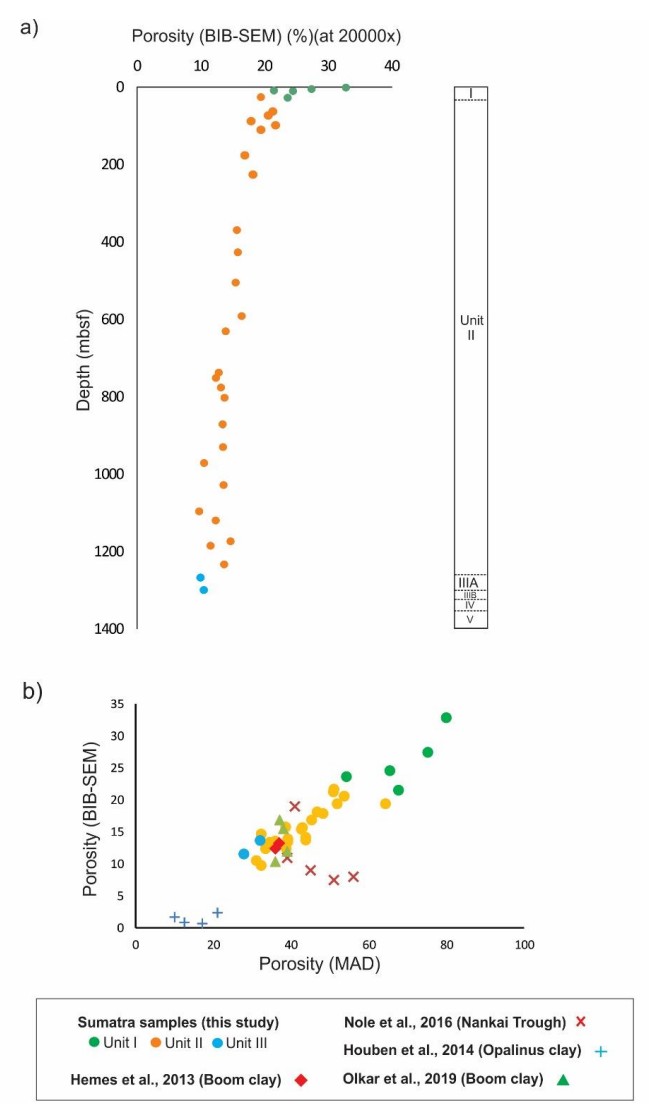


**Fig.3:** Porosity data for Units I (green dots), II (orange dots), and IIIA (blue dots). (a) BIB-SEM - depth plot. (b)
BIB-SEM porosity vs MAD porosity. Note: linear relationship that intersects origin. The data estimated by Hemes
et al., 2013; Houben et al., 2014; Olkar et al., 2019 also follow similar trend. However, the data estimated by Nole
et al., 2016 is deviated from the general trend.








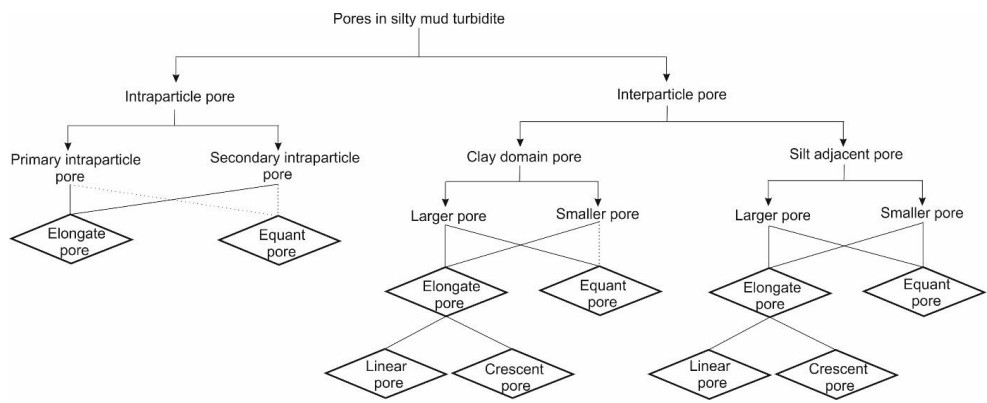



**Fig.4:** Classification scheme adopted to demonstrate pore reduction mechanics with increasing compactional
strain. Dashed lines indicate rare pore types.














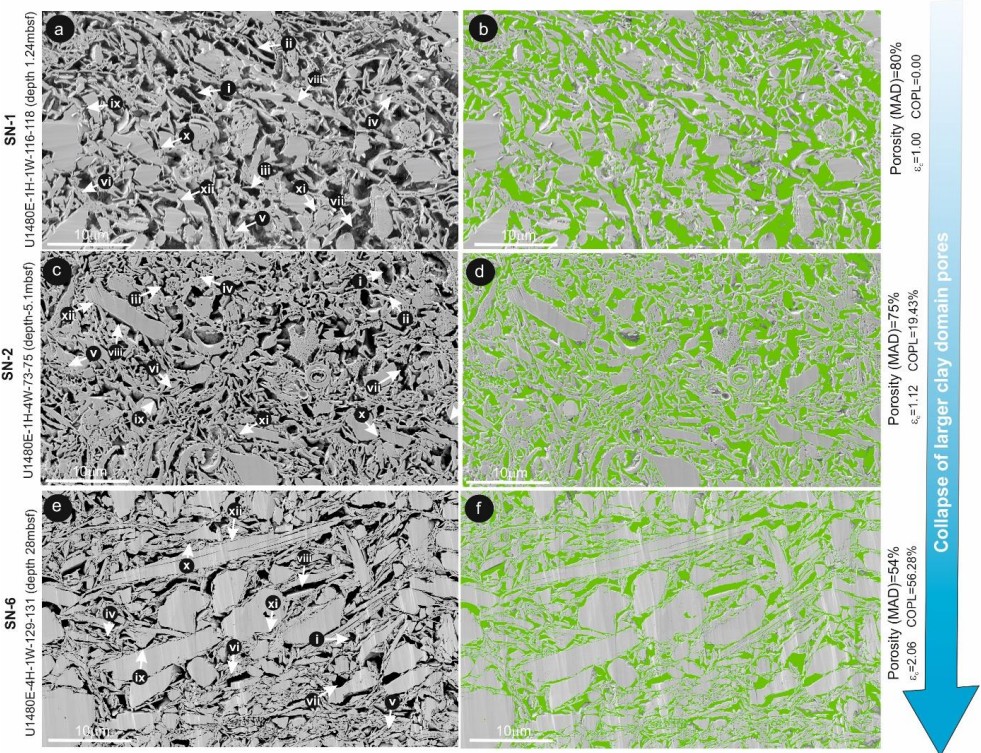

**Fig.5:** Microstructural overview (BIB-SEM) of samples SN-1 (a and b), SN-2 (c and d), and SN-6 (e and f). Green color represents segmented pores of the corresponding microstructure of sample.i = Equant large clay domain pores, ii = elongated large clay domain pores, iii = Crescent-shaped large clay domain pores, iv = equant small clay domain pores, v = Crescent-shaped small clay domain pores, vi = elongated small clay domain pores, vii = Equant large silt-adjacent pores, viii = elongated large silt-adjacent pores, ix = Crescent-shaped large silt-adjacent pores, x =equant small silt-adjacent pores, xi = Crescent-shaped small silt-adjacent pores, xii = elongated small silt-adjacent pores.

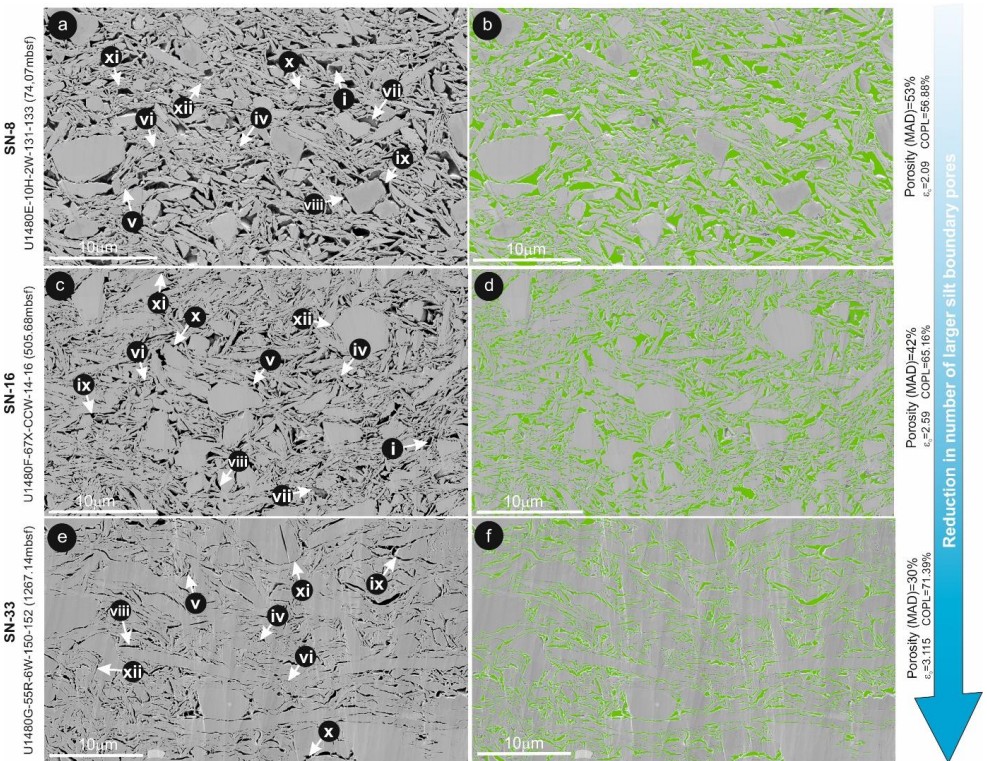

**Fig.6:** Microstructural overview (BIB-SEM) of samples SN-8 (a and b), SN-16 (c and d), and SN-32 (e and f). Green color represents segmented pores of the corresponding microstructure of sample. i = Equant large clay domain pores, ii = elongated large clay domain pores, iii = Crescent-shaped large clay domain pores, iv = equant small clay domain pores, v = Crescent-shaped small clay domain pores, vi = elongated small clay domain pores, vii = Equant large silt-adjacent pores, viii = elongated large silt-adjacent pores, ix = Crescent-shaped large silt-adjacent pores, x =equant small silt-adjacent pores, xi = Crescent-shaped small silt-adjacent pores, xii = elongated small silt-adjacent pores.


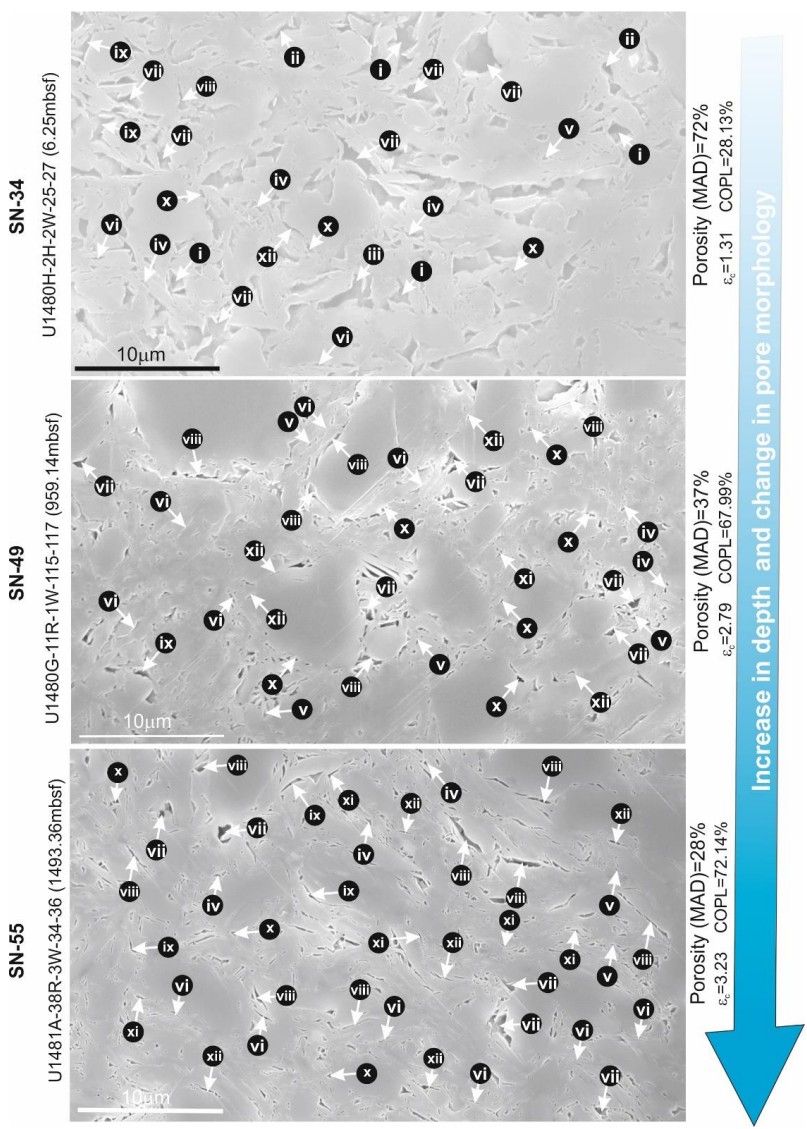

1074

Fig.7: Microstructural overview (Field Emission SEM) of samples SN-34, SN-49, and SN-55. i = Equant large clay domain pores, ii = elongated large clay domain pores, iii = Crescent-shaped large clay domain pores, iv = equant small clay domain pores, v = Crescent-shaped small clay domain pores, vi = elongated small clay domain pores, vii = Equant large silt-adjacent pores, viii = elongated large silt-adjacent pores, ix = Crescent-shaped large silt-adjacent pores, x =equant small silt-adjacent pores, xi = Crescent-shaped small silt-adjacent pores, xii = elongated small silt-adjacent pores.

1081





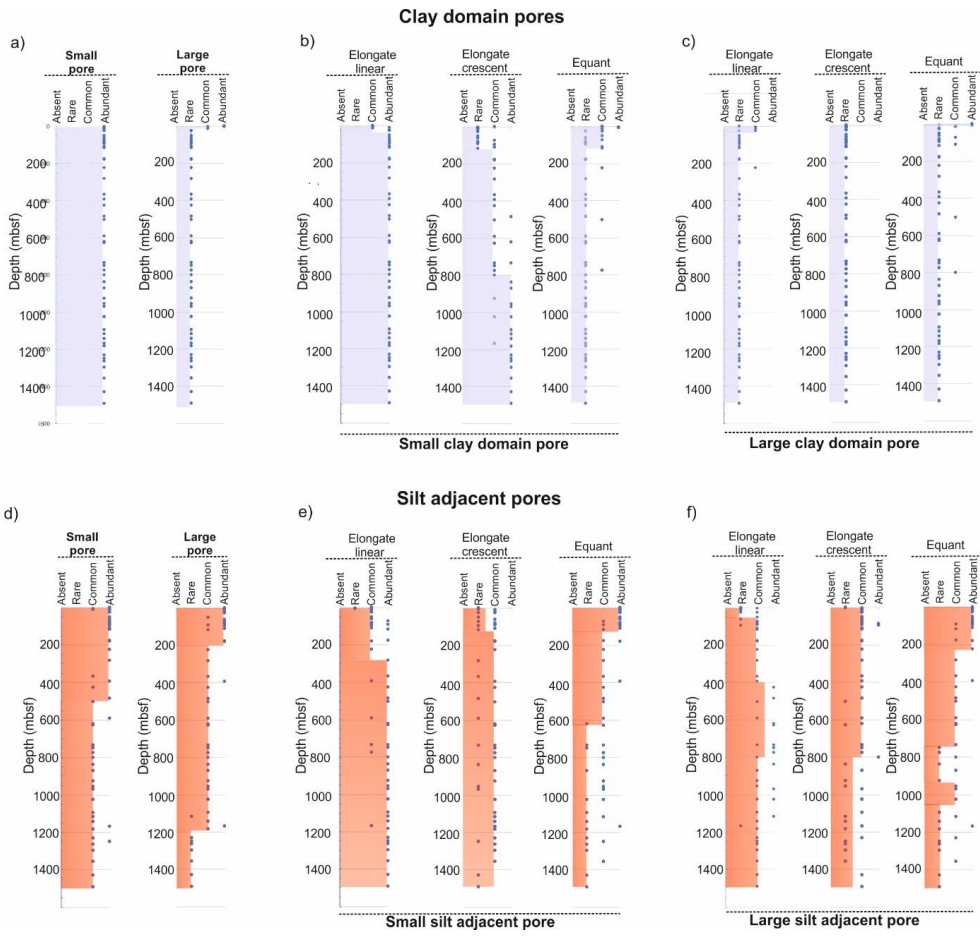

**Fig.8:** Pore type summary for clay domain (a-c) and silt-adjacent (d-f) pore types. (a) abundance of small and large clay domain pores; (b) and (c) depth progression of small and large clay domain pore morphologies; (d) abundance of small and large silt-adjacent pores; (e) and (f) depth progression of small and large clay domain pore morphologies. See text for abundance category definitions.

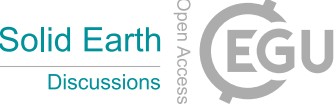



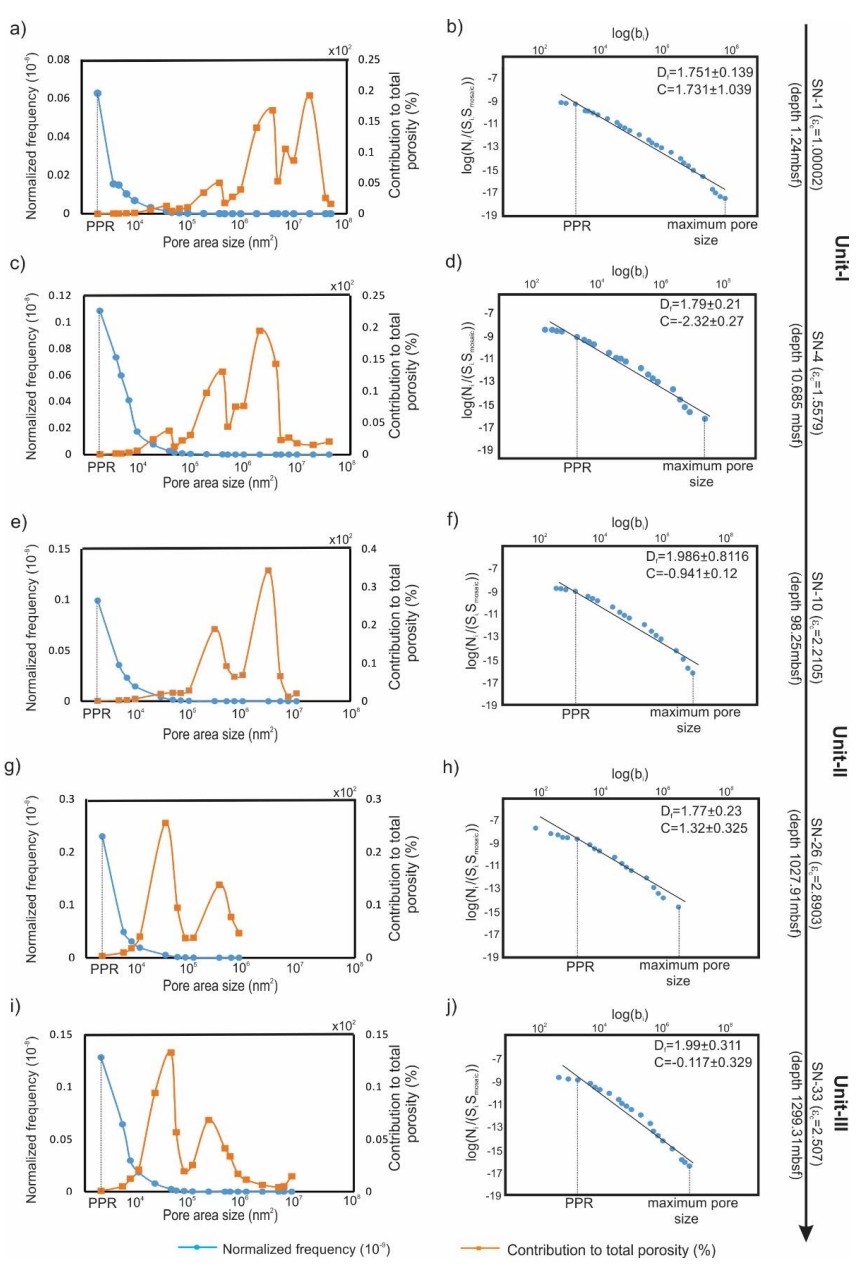

1093

**Fig.9:** Pore size distribution defined by pore area. Left column: normalized frequency (blue) and contribution to total porosity (orange). Right column: pore size-frequency log-log distribution. Power-law between PPR and maximum pore size interpreted as black line with corresponding regression parameters. Sample number, depth, and compactional strain defined along right side of diagram




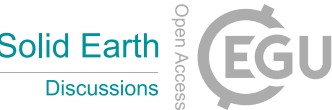

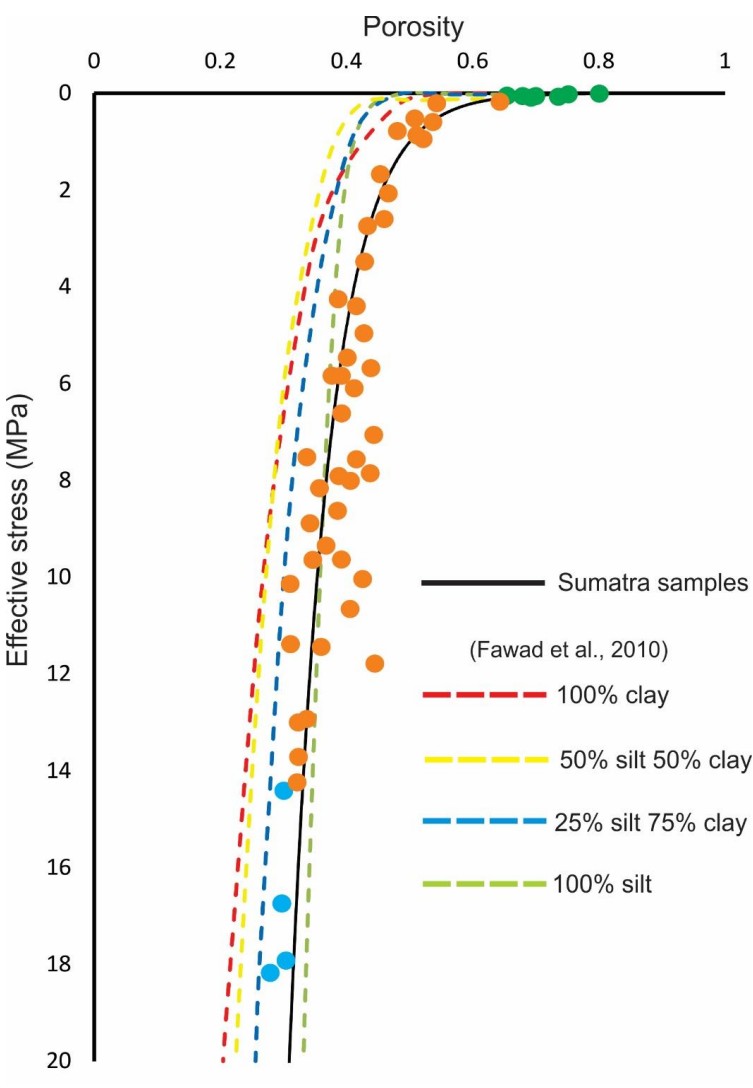


**Fig.10:** Effective stress vs porosity for experimental mixtures of clay and silt (dashed lines; Fawad et al., 2010)
compared with Sumatra data (Unit 1 = green; Unit II = orange; Unit III = blue). Solid black solid line is a best-
fit data regression for Sumatra samples.









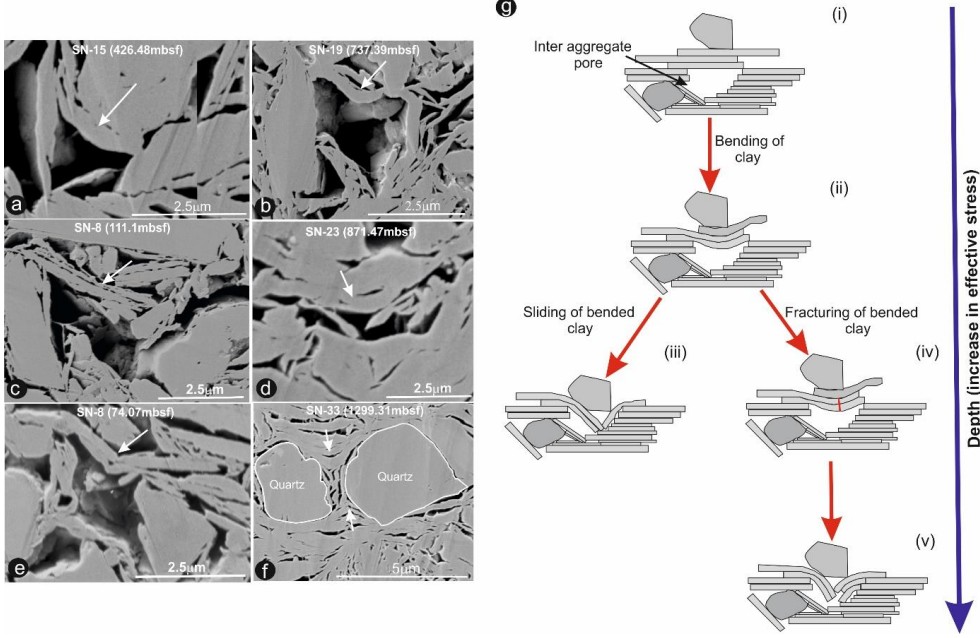


**Fig.11:** a-f: examples of bent clay particle on top of silt-adjacent larger pores; sample ID and depth labelled on photos. (g) Micromechanical model for collapse of large silt-adjacent pores.














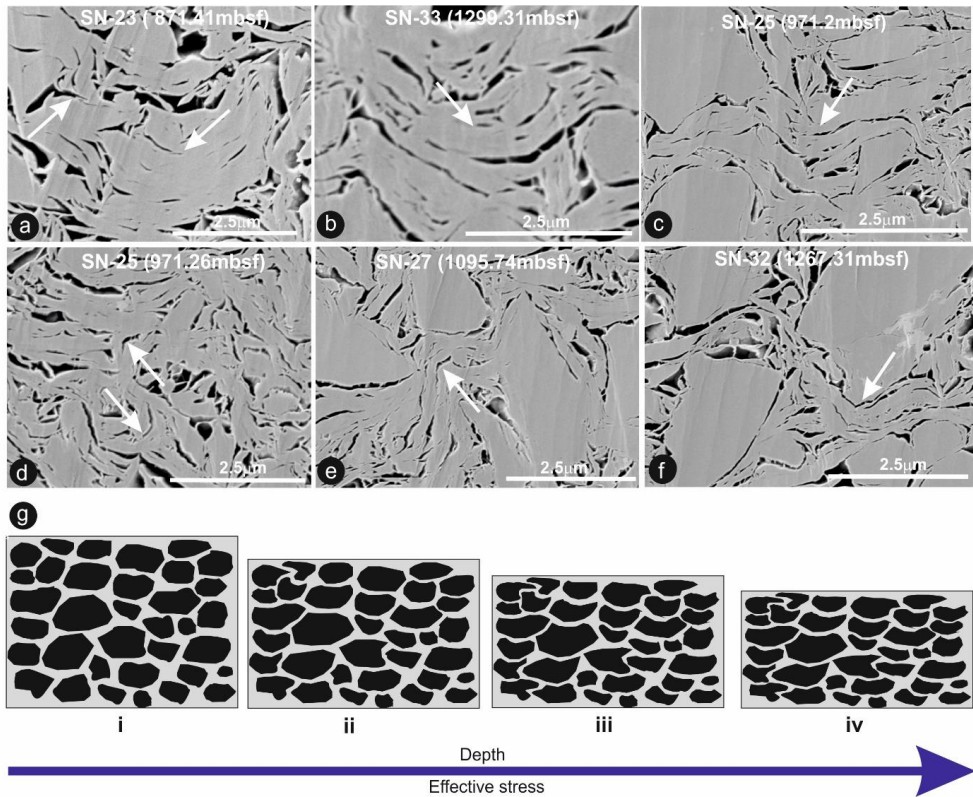

**Fig.12:** a-f: examples of clay aggregate bending (white and black arrows). (g) Conceptual diagram tracing porosity reduction and increase in preferred alignment of the long axes of pores by bending of clay perpendicular to applied vertical stress.



1133    Table 1: Clay mineralogy in subunits of the Sumatra succession

| Units | Smectite (%) | Illite (%) |
|---|---|---|
| Unit-I | 33 | 49 |
| Unit-II | 17 | 59 |
| Unit-IIIA | 73 | 19 |





Solid Earth Discussions — Open Access

**Table 2:** Summary of pore morphology evolution with depth. Abundant = >25% pores, common = 2%-25% pores, rare = 0-2% pores, absent = not observed.

| Depth (mbsf) | Sample no | Clay – Small pores ($<5\times10^5$ nm²) | Clay – Larger pores ($>5\times10^5$ nm²) | Clay Small pore – Elongate linear | Clay Small pore – Elongate crescent | Clay Small pore – Equant | Clay Larger pore – Elongate linear | Clay Larger pore – Elongate crescent | Clay Larger pore – Equant | Silt – Small pores ($<5\times10^5$ nm²) | Silt – Larger pores ($>5\times10^5$ nm²) | Silt Small pore – Elongate linear | Silt Small pore – Elongate crescent | Silt Small pore – Equant | Silt Larger pore – Elongate linear | Silt Larger pore – Elongate crescent | Silt Larger pore – Equant |
|---|---|---|---|---|---|---|---|---|---|---|---|---|---|---|---|---|---|
| 1.24 | SN-1 | Abundant | Rare | Abundant | Rare | Rare | Rare | Rare | Rare | Abundant | Abundant | Common | Rare | Abundant | Rare | Rare | Abundant |
| 5.10 | SN-2 | Abundant | Rare | Abundant | Rare | Common | Rare | Rare | Rare | Abundant | Abundant | Common | Rare | Abundant | Common | Common | Abundant |
| 6.25 | SN-34 | Abundant | Rare | Abundant | Rare | Common | Rare | Rare | Rare | Abundant | Abundant | Rare | Rare | Abundant | Rare | Rare | Abundant |
| 7.21 | SN-35 | Abundant | Rare | Abundant | Rare | Rare | Rare | Rare | Rare | Abundant | Abundant | Rare | Rare | Abundant | Rare | Common | Abundant |
| 9.18 | SN-3 | Abundant | Rare | Abundant | Common | Common | Rare | Rare | Common | Common | Abundant | Common | Rare | Abundant | Rare | Common | Abundant |
| 10.69 | SN-4 | Abundant | Rare | Abundant | Rare | Rare | Rare | Rare | Rare | Common | Abundant | Common | Common | Abundant | Rare | Common | Abundant |
| 14.28 | SN-36 | Abundant | Rare | Abundant | Rare | Rare | Rare | Rare | Rare | Abundant | Abundant | Common | Common | Abundant | Rare | Common | Abundant |
| 26.05 | SN-5 | Abundant | Rare | Abundant | Rare | Rare | Rare | Rare | Rare | Abundant | Abundant | Common | Common | Abundant | Rare | Common | Abundant |
| 28.00 | SN-6 | Abundant | Rare | Abundant | Common | Rare | Rare | Rare | Rare | Abundant | Abundant | Common | Rare | Abundant | Common | Common | Abundant |
| 28.12 | SN-37 | Abundant | Rare | Abundant | Common | Common | Rare | Rare | Common | Abundant | Abundant | Common | Common | Abundant | Common | Common | Abundant |
| 50.82 | SN-38 | Abundant | Rare | Abundant | Rare | Common | Rare | Rare | Rare | Abundant | Common | Common | Rare | Abundant | Common | Common | Abundant |
| 63.24 | SN-7 | Abundant | Rare | Abundant | Common | Rare | Rare | Rare | Rare | Abundant | Abundant | Common | Common | Abundant | Rare | Common | Abundant |
| 74.07 | SN-8 | Abundant | Rare | Abundant | Common | Rare | Common | Rare | Rare | Abundant | Abundant | Abundant | Rare | Common | Common | Common | Abundant |
| 83.02 | SN-39 | Abundant | Rare | Abundant | Common | Common | Common | Rare | Rare | Abundant | Abundant | Common | Common | Abundant | Common | Common | Abundant |
| 87.98 | SN-9 | Abundant | Rare | Abundant | Common | Rare | Rare | Rare | Rare | Abundant | Abundant | Common | Common | Abundant | Common | Common | Abundant |
| 92.82 | SN-41 | Abundant | Rare | Abundant | Common | Rare | Rare | Rare | Rare | Abundant | Common | Abundant | Common | Abundant | Common | Abundant | Abundant |
| 98.25 | SN-10 | Abundant | Rare | Abundant | Common | Rare | Rare | Rare | Rare | Abundant | Abundant | Common | Common | Common | Common | Common | Common |
| 111.10 | SN-11 | Abundant | Rare | Abundant | Common | Rare | Rare | Rare | Rare | Abundant | Abundant | Common | Rare | Abundant | Common | Abundant | Abundant |
| 117.13 | SN-40 | Abundant | Rare | Abundant | Abundant | Rare | Rare | Rare | Rare | Abundant | Common | Abundant | Rare | Common | Common | Common | Common |
| 176.50 | SN-12 | Abundant | Rare | Abundant | Common | Common | Rare | Rare | Common | Abundant | Abundant | Abundant | Common | Common | Common | Common | Common |
| 182.62 | SN-42 | Abundant | Rare | Abundant | Common | Rare | Rare | Rare | Rare | Abundant | Abundant | Common | Common | Abundant | Common | Common | Abundant |
| 226.70 | SN-13 | Abundant | Rare | Abundant | Abundant | Rare | Rare | Rare | Rare | Abundant | Common | Common | Common | Common | Common | Common | Abundant |
| 285.51 | SN-43 | Abundant | Rare | Abundant | Common | Rare | Rare | Rare | Rare | Abundant | Common | Abundant | Rare | Common | Common | Common | Common |
| 369.19 | SN-14 | Abundant | Rare | Abundant | Common | Rare | Rare | Rare | Rare | Common | Common | Abundant | Rare | Common | Common | Common | Common |
| 394.01 | SN-44 | Abundant | Rare | Abundant | Abundant | Rare | Rare | Rare | Rare | Abundant | Abundant | Common | Common | Abundant | Abundant | Common | Abundant |
| 426.68 | SN-15 | Abundant | Rare | Abundant | Common | Rare | Rare | Rare | Rare | Common | Common | Abundant | Common | Common | Abundant | Common | Common |
| 486.72 | SN-45 | Abundant | Rare | Abundant | Common | Common | Rare | Rare | Common | Abundant | Common | Abundant | Rare | Common | Abundant | Common | Common |
| 505.32 | SN-16 | Abundant | Rare | Abundant | Common | Rare | Rare | Rare | Common | Abundant | Common | Abundant | Common | Common | Common | Rare | Common |
| 592.42 | SN-17 | Abundant | Rare | Abundant | Abundant | Rare | Rare | Rare | Rare | Abundant | Common | Common | Rare | Common | Common | Common | Common |
| 621.2 | SN-46 | Abundant | Rare | Abundant | Abundant | Rare | Rare | Rare | Rare | Common | Common | Abundant | Common | Rare | Abundant | Common | Common |
| 630.55 | SN-18 | Abundant | Rare | Abundant | Common | Rare | Rare | Rare | Rare | Common | Common | Abundant | Common | Common | Abundant | Rare | Rare |
| 737.39 | SN-19 | Abundant | Rare | Abundant | Abundant | Rare | Rare | Rare | Rare | Common | Common | Common | Common | Rare | Common | Common | Common |
| 737.47 | SN-47 | Abundant | Rare | Abundant | Abundant | Rare | Rare | Rare | Rare | Common | Common | Abundant | Rare | Common | Abundant | Common | Common |
| 751.16 | SN-20 | Abundant | Rare | Abundant | Abundant | Rare | Rare | Rare | Rare | Common | Common | Abundant | Common | Rare | Abundant | Common | Rare |



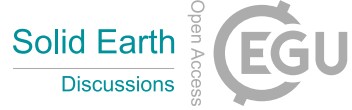

**Table 2:** Continued

| Depth (mbsf) | Sample no | Clay domain pores | | | | | | | | Silt adjacent pores | | | | | | | |
| --- | --- | --- | --- | --- | --- | --- | --- | --- | --- | --- | --- | --- | --- | --- | --- | --- | --- |
| | | Small pores (<5x10$^5$ nm$^2$) | Larger pores (>5x10$^5$ nm$^2$) | Small pore | | | Larger pore | | | Small pores (<5x10$^5$ nm$^2$) | Larger pores (>5x10$^5$ nm$^2$) | Small pore | | | Larger pore | | |
| | | | | Elongate linear | Elongate crescent | Equant | Elongate linear | Elongate crescent | Equant | | | Elongate linear | Elongate crescent | Equant | Elongate linear | Elongate crescent | Equant |
| 776.17 | SN-21 | Abundant | Rare | Abundant | Common | Rare | Rare | Rare | Rare | Common | Common | Common | Common | Common | Abundant | Common | Rare |
| 802.55 | SN-22 | Abundant | Rare | Abundant | Abundant | Rare | Rare | Rare | Rare | Common | Common | Abundant | Common | Common | Common | Abundant | Common |
| 841.56 | SN-48 | Abundant | Rare | Abundant | Abundant | Rare | Rare | Rare | Rare | Common | Common | Abundant | Rare | Common | Abundant | Rare | Rare |
| 871.87 | SN-23 | Abundant | Rare | Abundant | Abundant | Rare | Rare | Rare | Rare | Common | Common | Abundant | Common | Rare | Common | Common | Common |
| 929.81 | SN-24 | Abundant | Rare | Abundant | Common | Rare | Rare | Rare | Rare | Common | Common | Abundant | Common | Common | Common | Rare | Rare |
| 959.15 | SN-49 | Abundant | Rare | Abundant | Abundant | Rare | Rare | Rare | Rare | Common | Common | Abundant | Rare | Common | Common | Rare | Common |
| 971.26 | SN-25 | Abundant | Rare | Abundant | Abundant | Rare | Rare | Rare | Rare | Common | Common | Abundant | Rare | Common | Abundant | Common | Common |
| 1026.3 | SN-50 | Abundant | Rare | Abundant | Abundant | Rare | Rare | Rare | Rare | Common | Common | Abundant | Common | Common | Common | Common | Common |
| 1027.9 | SN-26 | Abundant | Rare | Abundant | Abundant | Rare | Rare | Rare | Rare | Common | Common | Abundant | Common | Rare | Abundant | Common | Common |
| 1095.7 | SN-27 | Abundant | Rare | Abundant | Abundant | Rare | Rare | Rare | Rare | Common | Common | Abundant | Common | Rare | Common | Common | Rare |
| 1119.7 | SN-28 | Abundant | Rare | Abundant | Abundant | Rare | Rare | Rare | Rare | Common | Rare | Abundant | Common | Common | Common | Rare | Common |
| 1145.9 | SN-51 | Abundant | Rare | Abundant | Abundant | Rare | Rare | Rare | Rare | Abundant | Common | Abundant | Common | Rare | Common | Rare | Rare |
| 1172.8 | SN-29 | Abundant | Rare | Abundant | Abundant | Rare | Rare | Rare | Rare | Common | Abundant | Abundant | Common | Abundant | Rare | Common | Abundant |
| 1184.3 | SN-30 | Abundant | Rare | Abundant | Rare | Rare | Rare | Rare | Rare | Common | Common | Abundant | Common | Rare | Common | Rare | Rare |
| 1233.1 | SN-31 | Abundant | Rare | Abundant | Rare | Common | Rare | Rare | Rare | Common | Rare | Abundant | Common | Rare | Common | Common | Common |
| 1251.5 | SN-52 | Abundant | Rare | Abundant | Rare | Common | Rare | Rare | Rare | Abundant | Rare | Abundant | Rare | Common | Common | Rare | Rare |
| 1267.1 | SN-32 | Abundant | Rare | Abundant | Rare | Rare | Rare | Rare | Rare | Common | Rare | Abundant | Common | Rare | Common | Rare | Rare |
| 1299.3 | SN-33 | Abundant | Rare | Abundant | Common | Common | Rare | Rare | Common | Common | Rare | Abundant | Common | Rare | Common | Rare | Rare |
| 1358.9 | SN-53 | Abundant | Rare | Abundant | Rare | Rare | Rare | Rare | Rare | Common | Rare | Abundant | Common | Common | Common | Rare | Rare |
| 1432.5 | SN-54 | Abundant | Rare | Abundant | Rare | Rare | Rare | Rare | Rare | Common | Rare | Abundant | Rare | Rare | Common | Common | Rare |





Table 3: Micromechanical particle deformation process with depth

| Depth | Sample | EE | EF | FF | Elongate particle bending | Cross particle fracture |
|---|---|---|---|---|---|---|
| 1.24 | SN-1 | Rare | Abundant | Common | Rare | Absent |
| 5.10 | SN-2 | Rare | Abundant | Common | Rare | Absent |
| 6.25 | SN-34 | Rare | Common | Abundant | Rare | Absent |
| 7.21 | SN-35 | Rare | Common | Abundant | Rare | Absent |
| 9.18 | SN-3 | Rare | Common | Abundant | Rare | Absent |
| 10.69 | SN-4 | Rare | Common | Abundant | Rare | Absent |
| 14.28 | SN-36 | Rare | Common | Abundant | Rare | Absent |
| 26.05 | SN-5 | Rare | Common | Abundant | Rare | Absent |
| 28.00 | SN-6 | Rare | Rare | Abundant | Rare | Absent |
| 28.12 | SN-37 | Rare | Common | Abundant | Rare | Rare |
| 50.82 | SN-38 | Rare | Rare | Abundant | Rare | Absent |
| 63.24 | SN-7 | Rare | Rare | Abundant | Rare | Absent |
| 74.07 | SN-8 | Rare | Rare | Abundant | Rare | Absent |
| 83.02 | SN-39 | Rare | Rare | Abundant | Rare | Absent |
| 87.98 | SN-9 | Rare | Rare | Abundant | Rare | Absent |
| 92.82 | SN-41 | Rare | Rare | Abundant | Rare | Absent |
| 98.25 | SN-10 | Rare | Common | Abundant | Common | Absent |
| 111.10 | SN-11 | Rare | Rare | Abundant | Common | Rare |
| 117.13 | SN-40 | Rare | Rare | Abundant | Rare | Absent |
| 176.50 | SN-12 | Rare | Rare | Abundant | Rare | Absent |
| 182.62 | SN-42 | Rare | Rare | Abundant | Common | Absent |
| 226.70 | SN-13 | Rare | Rare | Abundant | Common | Absent |
| 285.51 | SN-43 | Rare | Rare | Abundant | Rare | Absent |
| 369.19 | SN-14 | Rare | Rare | Abundant | Common | Absent |
| 394.01 | SN-44 | Rare | Rare | Abundant | Common | Absent |
| 426.68 | SN-15 | Rare | Rare | Abundant | Common | Absent |
| 486.72 | SN-45 | Rare | Rare | Abundant | Rare | Absent |
| 505.32 | SN-16 | Rare | Rare | Abundant | Common | Absent |
| 592.42 | SN-17 | Rare | Rare | Abundant | Common | Absent |
| 621.2 | SN-46 | Rare | Rare | Abundant | Rare | Absent |
| 630.55 | SN-18 | Rare | Common | Abundant | Common | Rare |
| 737.39 | SN-19 | Rare | Rare | Abundant | Rare | Rare |
| 737.47 | SN-47 | Rare | Rare | Abundant | Rare | Absent |
| 751.16 | SN-20 | Rare | Rare | Abundant | Common | Absent |
| 776.17 | SN-21 | Rare | Rare | Abundant | Common | Absent |
| 802.55 | SN-22 | Rare | Rare | Abundant | Rare | Absent |
| 841.56 | SN-48 | Rare | Rare | Abundant | Common | Absent |
| 871.87 | SN-23 | Rare | Rare | Abundant | Common | Rare |
| 929.81 | SN-24 | Rare | Rare | Abundant | Rare | Rare |
| 959.15 | SN-49 | Rare | Rare | Abundant | Rare | Rare |
| 971.26 | SN-25 | Rare | Rare | Abundant | Common | Rare |
| 1026.3 | SN-50 | Rare | Rare | Abundant | Common | Absent |
| 1027.9 | SN-26 | Rare | Rare | Abundant | Common | Absent |
| 1095.7 | SN-27 | Rare | Rare | Abundant | Common | Absent |
| 1119.7 | SN-28 | Rare | Rare | Abundant | Common | Rare |
| 1145.9 | SN-51 | Rare | Rare | Abundant | Rare | Absent |
| 1172.8 | SN-29 | Rare | Rare | Abundant | Common | Rare |
| 1184.3 | SN-30 | Rare | Rare | Abundant | Common | Absent |
| 1233.1 | SN-31 | Rare | Rare | Abundant | Common | Rare |
| 1251.5 | SN-52 | Rare | Rare | Abundant | Rare | Absent |
| 1267.1 | SN-32 | Rare | Rare | Abundant | Abundant | Rare |
| 1299.3 | SN-33 | Rare | Rare | Abundant | Abundant | Absent |
| 1358.9 | SN-53 | Rare | Rare | Abundant | Common | Absent |
| 1432.5 | SN-54 | Rare | Rare | Abundant | Common | Absent |