# Peer review of "Mechanical compaction mechanisms in the input sediments of the"

_Solid Earth, 2022_

## Referee Comment (RC1)

[revised manuscript text omitted]

*[handwritten annotation: In these plots, are the D values calculated from all points or just the points that avoid the artificial tail-offs at both ends of the line?]*

$$\log\left(\frac{N_i}{b_i S_{mosaic}}\right) = -D.\log(S_{pore}) + Log\ C \qquad \text{(Eqn-4)}$$

*[handwritten annotation: Ni; bi]*

Where $N_i$= number of pores with area $S_{pore}$, $b_i$= bin size, $S_{mosaic}$= surface area of the current mosaic,

C=constant, and D= power-law exponent. The resulting power-law exponent (D) varies between 1.70 to 2.00

(Supplementary data-7).

**Effect of texture on porosity, pore morphology, and orientation of pores**

We analyzed six samples (SN-7, SN-9, SN-17, SN-28, SN-29 and SN-31) that are enriched in silt content compared to the rest of the mud samples (Supplementary data-7). Silt content has a positive correlation to the total SEM

porosity. For example, sample SN-29 (1172.88 mbsf) exhibits a BIB-SEM porosity of 14% whereas other samples from a similar depth with less silt exhibit an average BIB-SEM porosity of 12% (Supplementary data-7) at 20000x magnification. The samples with greater silt content are also enriched in equant-shaped silt-adjacent larger pores (Supplementary data 13a). We also estimated the orientation of the long axis of pores for these three samples and plotted the obtained results as rose diagrams (Supplementary data 13b). The obtained results exhibit a relatively weak preferred alignment of the long axis of pores with respect to the bedding planes (Supplementary data 11b).

**Discussions**

**Effective stress vs porosity: A comparison with experimental study**

To understand the consolidation mechanisms of the Sumatra sediment, we estimated vertical effective stress following the steps proposed by Hüpers et al., 2015. Following Terzaghi and Peck (1948) vertical effective stress ($\sigma_v{}'$)

is expressed as:

$$\sigma_v{}' = \sigma_v - P_f \qquad \text{(Eqn-7)}$$

Here $\sigma_v$ = total vertical stress caused by the overburden load, and $P_f$= fluid pressure. To compute vertical effective stress of a layered sediment, we use Eqn 8:

$$\sigma_v{}' = \sum(\rho_s - \rho_w).g.\Delta z \qquad \text{(Eqn-8)}$$

[revised manuscript text omitted]

Not convinced tables 2 + 3 add anything. Supplementary Data? would prefer to see better morphology data, either tabula... [cut off] or graphically or both... [cut off]

**Table 2:** Summary of pore morphology evolution with depth. Abundant = >25% pores, common = 2%-25% pores, rare = 0-2% pores, absent = not observed.

| Depth (mbsf) | Sample no | Clay domain pores | | | | | | | | Silt adjacent pores | | | | | | | |
| --- | --- | --- | --- | --- | --- | --- | --- | --- | --- | --- | --- | --- | --- | --- | --- | --- | --- |
| | | Small pores (<5x10⁵ nm²) | Larger pores (>5x10⁵ nm²) | Small pore Elongate linear | Small pore Elongate crescent | Small pore Equant | Larger pore Elongate linear | Larger pore Elongate crescent | Larger pore Equant | Small pores (<5x10⁵ nm²) | Larger pores (>5x10⁵ nm²) | Small pore Elongate linear | Small pore Elongate crescent | Small pore Equant | Larger pore Elongat elinear | Larger pore Elongate crescent | Larger pore Equant |
| 1.24 | SN-1 | Abundant | Rare | Abundant | Rare | Rare | Rare | Rare | Rare | Abundant | Abundant | Common | Rare | Abundant | Rare | Rare | Abundant |
| 5.10 | SN-2 | Abundant | Rare | Abundant | Rare | Common | Rare | Rare | Rare | Abundant | Abundant | Common | Rare | Abundant | Common | Common | Abundant |
| 6.25 | SN-34 | Abundant | Rare | Abundant | Rare | Common | Rare | Rare | Rare | Abundant | Abundant | Rare | Rare | Abundant | Rare | Rare | Abundant |
| 7.21 | SN-35 | Abundant | Rare | Abundant | Rare | Rare | Rare | Rare | Rare | Abundant | Abundant | Rare | Rare | Abundant | Rare | Common | Abundant |
| 9.18 | SN-3 | Abundant | Rare | Abundant | Common | Common | Rare | Rare | Common | Common | Abundant | Common | Common | Abundant | Rare | Common | Abundant |
| 10.69 | SN-4 | Abundant | Rare | Abundant | Rare | Rare | Rare | Rare | Rare | Common | Abundant | Common | Common | Abundant | Rare | Common | Abundant |
| 14.28 | SN-36 | Abundant | Rare | Abundant | Rare | Rare | Rare | Rare | Rare | Abundant | Abundant | Common | Common | Abundant | Rare | Common | Abundant |
| 26.05 | SN-5 | Abundant | Rare | Abundant | Rare | Rare | Rare | Rare | Rare | Abundant | Abundant | Common | Rare | Abundant | Rare | Common | Abundant |
| 28.00 | SN-6 | Abundant | Rare | Abundant | Common | Rare | Rare | Rare | Rare | Abundant | Abundant | Common | Common | Abundant | Rare | Common | Abundant |
| 28.12 | SN-37 | Abundant | Rare | Abundant | Common | Common | Rare | Rare | Common | Abundant | Abundant | Common | Rare | Abundant | Common | Common | Abundant |
| 50.82 | SN-38 | Abundant | Rare | Abundant | Rare | Common | Rare | Rare | Rare | Abundant | Common | Common | Rare | Abundant | Common | Common | Abundant |
| 63.24 | SN-7 | Abundant | Rare | Abundant | Common | Rare | Rare | Rare | Rare | Abundant | Abundant | Common | Common | Abundant | Rare | Common | Abundant |
| 74.07 | SN-8 | Abundant | Rare | Abundant | Common | Rare | Rare | Rare | Rare | Abundant | Abundant | Rare | Rare | Abundant | Common | Common | Abundant |
| 83.02 | SN-39 | Abundant | Rare | Abundant | Common | Common | Common | Rare | Common | Abundant | Abundant | Common | Common | Abundant | Common | Abundant | Abundant |
| 87.98 | SN-9 | Abundant | Rare | Abundant | Common | Rare | Common | Rare | Rare | Common | Abundant | Common | Rare | Common | Common | Common | Common |
| 92.82 | SN-41 | Abundant | Rare | Abundant | Common | Rare | Abundant | Rare | Rare | Abundant | Abundant | Abundant | Rare | Abundant | Common | Abundant | Abundant |
| 98.25 | SN-10 | Abundant | Rare | Abundant | Common | Rare | Common | Rare | Rare | Abundant | Common | Common | Common | Common | Rare | Common | Common |
| 111.10 | SN-11 | Abundant | Rare | Abundant | Common | Rare | Common | Rare | Rare | Abundant | Abundant | Common | Rare | Abundant | Common | Common | Abundant |
| 176.13 | SN-40 | Abundant | Rare | Abundant | Abundant | Rare | Abundant | Rare | Rare | Abundant | Common | Abundant | Common | Common | Common | Common | Common |
| 176.50 | SN-12 | Abundant | Rare | Abundant | Common | Common | Abundant | Rare | Common | Abundant | Abundant | Common | Rare | Abundant | Common | Abundant | Abundant |
| 182.62 | SN-42 | Abundant | Rare | Abundant | Common | Rare | Common | Rare | Rare | Abundant | Abundant | Common | Common | Abundant | Common | Common | Common |
| 226.70 | SN-13 | Abundant | Rare | Abundant | Abundant | Rare | Abundant | Rare | Common | Abundant | Abundant | Abundant | Common | Common | Common | Common | Abundant |
| 285.51 | SN-43 | Abundant | Rare | Abundant | Common | Rare | Common | Rare | Rare | Abundant | Common | Abundant | Common | Rare | Common | Common | Common |
| 369.19 | SN-14 | Abundant | Rare | Abundant | Common | Rare | Common | Rare | Rare | Abundant | Common | Common | Common | Common | Common | Common | Common |
| 394.01 | SN-44 | Abundant | Rare | Abundant | Abundant | Rare | Abundant | Rare | Rare | Abundant | Abundant | Abundant | Rare | Abundant | Common | Common | Abundant |
| 426.68 | SN-15 | Abundant | Rare | Abundant | Common | Rare | Common | Rare | Rare | Common | Common | Abundant | Common | Common | Abundant | Common | Common |
| 486.72 | SN-45 | Abundant | Rare | Abundant | Common | Common | Abundant | Rare | Common | Abundant | Abundant | Abundant | Common | Common | Abundant | Common | Common |
| 505.32 | SN-16 | Abundant | Rare | Abundant | Common | Rare | Common | Rare | Rare | Abundant | Common | Abundant | Rare | Rare | Common | Common | Common |
| 592.42 | SN-17 | Abundant | Rare | Abundant | Abundant | Common | Common | Rare | Common | Common | Common | Common | Common | Common | Common | Rare | Common |
| 621.2 | SN-46 | Abundant | Rare | Abundant | Common | Rare | Abundant | Rare | Rare | Abundant | Common | Abundant | Common | Common | Abundant | Common | Common |
| 630.55 | SN-18 | Abundant | Rare | Abundant | Abundant | Rare | Abundant | Rare | Rare | Common | Common | Abundant | Common | Rare | Abundant | Rare | Common |
| 737.39 | SN-19 | Abundant | Rare | Abundant | Abundant | Rare | Common | Rare | Rare | Common | Common | Abundant | Common | Common | Common | Rare | Common |
| 737.47 | SN-47 | Abundant | Rare | Abundant | Abundant | Rare | Abundant | Rare | Rare | Common | Common | Common | Rare | Rare | Abundant | Common | Common |
| 751.16 | SN-20 | Abundant | Rare | Abundant | Abundant | Rare | Abundant | Rare | Rare | Common | Common | Abundant | Common | Common | Common | Common | Common |
| 776.17 | SN-21 | Abundant | Rare | Abundant | Common | Common | Rare | Rare | Rare | Common | Common | Common | Common | Rare | Abundant | Common | Rare |
| 802.55 | SN-22 | Abundant | Rare | Abundant | Abundant | Rare | Abundant | Rare | Rare | Common | Common | Common | Common | Common | Common | Common | Common |
| 841.56 | SN-48 | Abundant | Rare | Abundant | Abundant | Rare | Abundant | Rare | Rare | Common | Common | Abundant | Rare | Common | Abundant | Abundant | Common |
| 871.87 | SN-23 | Abundant | Rare | Abundant | Abundant | Rare | Abundant | Rare | Rare | Common | Common | Abundant | Common | Rare | Common | Common | Common |

---

## Referee Comment (RC2)

[referee-annotated manuscript omitted]

---

## Author Comment (AC1)

To,

Prof. Virginia Toy,

Editor, Solid earth

*Sub: Submission of Revised Manuscript (SE-2022-11)*

Dear Prof. Toy,

Thank you for the review reports on our manuscript entitled "**Mechanical compaction mechanisms in the input sediments of the Sumatra Subduction Complex- insights from microstructural analysis of cores from IODP Expedition- 362**" written by Lahiri, S.; Milliken, K. L.; Vrolijk, P., Desbois, G.; and Urai, J. L. "Major Revision" was recommended based on the review report. Reviewer#1 (Dr. Dave Dewhurst) stated regarding the manuscript that, 'this is a really an interesting manuscript with some fabulous SEM images of compacting clays and great insights into processes occurring during early mechanical compaction'. Reviewer#2 (Dr. Bernhard Schuck) commented that "the manuscript is well-written and address a topic relevant to solid earth". However, the reviewers also raised several questions regarding clay mineralogy, bulk rock mineralogy, drying artefacts etc.

We appreciate the constructive review by the reviewers, and in the revised manuscript we have addressed all the issues raised by them. ***In the revised manuscript, all the changes based on suggestions of the reviewers are highlighted in green color (reviewer#1, Dave Dewhurst) and yellow color (reviewer#2, Bernhard Schuck).***

I hope that the revised version will be accepted for publication in the *Solid Earth*.

Thanking you,

Yours sincerely

Sivaji Lahiri

(For the authors)

**Reviewer 1# (Dr. Dave Dewhurst)**

**Comment 1:** My main concern with this manuscript is that the description of the clay mineralogy of the sediments is somewhat confusing and this requires significant clarification by the authors. The paragraph on page 4, line 113-121 notes a clay mineral assemblage of "illite with minor smectite and chlorite", noting 33% smectite in unit 1 and 73% in unit III (Table 1), numbers which are not really minor (and are averages really useful in such thick units?). The illite percentages are then given followed by a discussion of illite-smectite (I-S) expandability. However, I-S is not documented at all in table 1, just presumably pure illite and smectite. How much mixed-layer clay is present, and is it being properly differentiated from the pure end members? What is the I-S composition? This is actually important for the whole manuscript since porosity corrections are required for smectite-rich sediments and this may impact significantly on results, especially as there seems to be 15-20% "missing porosity" as noted in the discussion between MAD and SEM porosity (although there are other possibilities for this). This lack of clarity over clay mineralogy also comes back in the discussion (p12, L385-388), where the numbers quoted in the text are not consistent with table 1. For me, I feel it would be really informative to have the clay compositions plotted up against depth and added to figure 2 for comparison. I'm sure IODP have the data and although the measurements wouldn't be from exactly the same points as measured here, they would still provide clear trend information.

*Response: We agree about the need for greater clarity on clay mineralogy. In the revised manuscript, we have modified Fig.2 and Table-1. In Fig.2, we added Smectite content vs depth, and Illite content vs depth plots for both drill holes (U1480 and U1481) samples separately. In Table-1, we added separate sections for U1480 and U1481 drill hole, and also mentioned average Smectite, Illite and I/S expandability for each Unit. In addition, we also added a new supplementary data (supplementary data-3) to depict variation of I/S expandability with depth. Nevertheless, to clarify this particular issue we have also modified the section 'Geological Background and drilling' and provided mineralogical description in details (Line number: 131 to 145). I hope, these modifications resolve this issue.*

**Comment 2:** P2, L52-54, depositional environment is also a strong control on porosity evolution, compaction and diagenesis in mudrocks (e.g. Baruch et al., 2015, AAPG; Delle Piane et al 2015 MPG) as the initial clay and rigid grain compositions significantly affect both compaction (as this manuscript shows) and subsequent diagenetic alteration due to variations in composition.

*Response: A line is added in the main text (Line number: 56 to 59).*

**Comment 3:** Somewhere in the geological background and drilling section, there should probably be a mention of whether these sediments have been uplifted at all or are currently at maximum burial depth. I would assume the latter just from how things have been written but probably best to have a clear statement.

*Response: A line is added in the text to clarify this issue (Line number: 148 to 149).*

**Comment 4:** L137-147, somewhere in this paragraph there needs to be a description of how the samples were freeze dried as this is absolutely critical in very soft clay bearing sediments when trying to get quantitative data from pore characterisation techniques, whether that be via SEM or gas/mercury porosimetry. Need to know things like the method employed, temperatures achieved, how long they were frozen for before drying, how long the drying took etc etc. This goes to understanding the quality of results and I'm a little surprised there is no discussion of this. There was work done on this going back to the early 1980s looking at fabrics and microstructure of soft clays and the impact drying can have on porosity measurements and particle preferred orientation. There should be some consideration of this work and later work for Example in the Microstructure of Fine-Grained Sediments compilation quoted in the Bennett et al reference in the manuscript. Given the "missing porosity" between MAD and SEM quoted later in the paper, could the drying methods account for that? This is a really important topic where the authors need to convince the reader that the data quality is good. A few of the early references may prove helpful and these (plus younger ones that could be found through citations) could help formulate the discussion further. I'll come back to this point below.

*Response: For Aachen sample sets, we have added a section in the main text describing the condition maintained for drying the samples at IODP repository, Japan (line number: 168 to 171). For BEG sample set, the drying condition and after drying effect is described at line number: 223 to 225.*

*We agree that drying artefacts can significantly affect the MAD porosity vs BIB-SEM porosity plot. We are grateful to the reviewer for identifying this problem in our old version of the manuscript. In the revised version, we have added a section in the discussion describing this aspect to clarify this issue (Line number: 459 to 472).*

**Comment 5:** P5, L169, The PPR and REA section needs some discussions around the assumptions, errors and limitations of determining porosity and pore sizes from 2D images. Pores are non-spherical 3D objects that are cut through in random orientations and at random points in a 2D thin section, so measurements taken on the images have associated assumptions and errors.

*Response: This is clarified in the main text (Line number: 209 to 218).*

**Comment 6:** P6, L195, introduced the shipboard MAD measurements in the results section. Again, even though done elsewhere, a brief description of the methodology is warranted in the section above as to how this was performed and then point the reader to the appropriate reference for further details. These data are absolutely critical to the manuscript, so the details are sorely needed. In addition, Figure 2 shows some data in yellow points as measured at Aachen and BEG; were these done by the same method as shipboard or different? Again, a description is warranted.

*Response: DONE. A brief description of the methodology employed for generating MAD data set is added in the main text (238 to 242).*

*Blue points are describing Shipboard MAD (Moisture and density) porosity vs depth data for mudstone samples recovered from Sites U1480 and U1481. Yellow points are describing analysed samples at Aachen and BEG. To bring clarity, we modified the main text (Line number:162 to 165) and also caption of Fig.2.*

**Comment 7:** L238-239, the correlation coefficient between the two porosity measurements should be shown either in the text or on Figure 3b.

*Response**: DONE. (Fig.3b and line number: 288 in the main text).*

**Comment 8:** P11, L341, in these plots it should be stated which points are used to calculate the D values. Is it all points or just the points that avoid the artificial tail offs from sampling bias/resolution issues at either end of the lines? Probably should be the latter but some of the trend lines do not lie through the straight-line section of the points.

*Response: Yes, we have considered the points that avoid artificial tail off due to resolution issues. In the diagram we plotted the line corresponding to the average 'D' value (i.e. the average of highest and lowest estimated 'D' values). However, we have again rechecked each plot and corresponding D value for each sample.*

Comment 9: For me, the whole discussion is a bit weak and lacking in detail. There are some interesting points made and comparisons, but they are not really built on very much and tend to be almost standalone points based on a couple of sentences. There also seems to be potential background discussion items that probably should be in there but are currently not referred to. For example, the soil mechanics community has done a lot of work around natural clays and remoulded clays (similar to the Mondol/Fawad type work) but this is not even discussed although thoroughly relevant to the behaviour noted here (e.g. classic papers by Burland, 1990, Geotechnique; I think also Lupini et al 1991, Geotechnique, which is currently quoted in the reference section but isn't in the text). The references I note above in point 4, although old, also discuss topics alluded to here in terms of pore collapse with compaction, the very topic of the paper. Perhaps a bit more background in the introductory sections would provide a good lead into the discussion of the results obtained. Supplement 15 should also be in the discussion as it provides some details that are sorely missing at the moment.

*Response: We are thankful to the reviewer for suggesting this. In the revised version of the manuscript, we have modified and improved a section in the introduction (Line number:75 to 93) to discuss previous studies, and also subsequently improved discussion section (line number: 543 to 561). In addition, we have also reviewed recent literatures on mechanical compaction and added a new supplementary data-1 to describe these studies in details.*

*Supplementary data-15 (according to old version) is added to the main text (line number:501 to 522)*

**Comment 10:** P12, L400-405, it would be better to show the supplementary data 14 in the text rather than outside the manuscript. This is actually a pretty important result and consistent with the topic of the paper.

*Response: DONE. We have incorporated it in the main text.*

**Comment 11:** P13, L415-417, this is a good point here about the smectite content potentially resulting in over-estimated MAD porosity and one of the things I was getting at above in point 1. However, until there is some better clarity about the clay mineral composition, it is a moot point. I suspect this is a contributor for sure but again this needs a more detailed discussion as to why this is the case and appropriate supporting references quoted.

*Response: We have modified Fig.2 for clarification on the clay mineralogy (details mentioned in the response of Comment 1). We also clarified this issue in the text (Line number: 473 to 477).*

**Comment 12:** P15, L506, the authors discuss "intra-crystalline" slip as a mechanism for particle re-arrangements during compaction. This term has a very specific meaning, usually around things like dislocations and the like within crystals, calcite being a classic example at relatively low temperatures and pressures. One thing the authors haven't really discussed is the flocculation and aggregation of clay minerals (although flocculation first makes an appearance in the conclusions section on P17, L545 and aggregate in the discussion). The results of these processes can be clearly seen in the SEM images the authors show, often with multiple clay platelets stuck together in aggregations, but this is not described in the images clearly. The intra-crystalline slip quoted here appears to be more like intra-aggregate slip of particles or inter-platelet slip and these would be better ways to describe these phenomena. Again, there is a lot of background work done on clay fabrics in compacting clays with well established terminology which isn't referenced here, I think some in the references I have noted above for the early work and plenty in the Microstructure of Fine-Grained Sediments compilation as starting points. The manuscript would really benefit from a brief introduction to typical microstructural fabrics of clays which then naturally leads into the discussion.

*Response: We are aware of the important role assigned to flocculation and other mechanisms of particle aggregation in clay-particle deposition. The image regions we have termed "clay-rich domains" appear as somewhat discrete masses because they are distributed between the larger silt particles, but there is no clear way to ascertain if particular particles within the observed clay-rich domains were ever joined together as a discrete floc prior to deposition. We have endeavored in the revision to be consistent in using the less interpretive term "clay particle" or "clay-rich domain" and to generally avoid the term "aggregate" or any reference to flocculated structure, as our focus is on the post-depositional modification of the pore structure, regardless of the depositional mechanism. The terminology intra-crystalline slip is changed to 'Intra-particle slip (Line number: 27 and 605 of revised manuscript)*

**Comment 13:** As an overall comment, there is a lot of good data in the supplementary data that would really back up the observations made in the text. I have already mentioned 14 above, but 11 and 12 to me are also no brainers for supporting the hypothesis put forward by the authors. To be honest, I don't think tables 2 and 3 bring anything to the paper, they are not things that can be quantified, and these for sure should be supplementary data as figure 8 covers their input adequately. If one was going to include a table of useful data in the paper, I would pick supplement 7, perhaps a slimmed down version with some columns removed and the full table still provided as a supplement. This would provide much more useful and quantifiable data to the reader.

*Response: We have added Supplementary data-14 to the main text (Fig.12 in revised manuscript). We deleted Supplementary data-11 (number from old version).*

*We have removed table-2 and 3 (number is from old version of manuscript) from the main text. We added a slimmed version of Supplementary data-7 (number from old version) to the main text as Table-2.*

**Comment 14:** Following on with the discussion on supplementary data, the large pore outlined in the image of the deepest sample in supplement 10 looks like a plucked grain to me. One can see the compacted clays around the edge of the hole outline and within the hole. I would use a better image than this if you have one. The other two images in this supplement for the shallower samples are clearly geologically realistic and relate to the interactions between rigid grains.

*Response: DONE. Now we have modified Supplementary data 10 (according to old version of manuscript) with a new image for the deepest sample. This is represented in Supplementary data-14 in revised version.*

**Comment 15:** For figure 1b, the abbreviation HANP needs spelling out or removing if not relevant to the study.

*Response: DONE. HANP is removed from Figure 1b.*

**Comment 16:** In figure 2, b and c are the wrong way round. This plot would benefit from clay mineralogy data (e.g. illite vs depth, smectite vs depth, clay content vs depth, I-S vs depth etc) on a single plot (or added to the porosity plots).

*Response: DONE. We have modified Fig.2 and incorporated depth vs Smectite, and depth vs illite for all samples. Moreover, we also added a new Supplementary data-3 for I/S expandability vs depth for the analysed samples.*

**Comment 17:** Figure 3 would benefit from having a 1:1 line for easy reference for the reader. Also, the Boom Clay symbol colour on the graph doesn't match the legend.

*Response: Done*

**Comment 18:** The references are a real mess, many not in the text (all the ones with red ticks on the marked manuscript were ones I could find, including in all the supplementary data), many dates are inconsistent between text and reference list (marked in manuscript) and many in the text but not in the reference list. To be honest, this is much more than a reviewer should have to deal with and tidy up!

*Response: Done.*

**Comment 19:** There are a bunch of typos and suggested edits also in the marked-up manuscript supplied.

*Response: DONE.*

**Reviewer2# (Dr. Bernhard Schuck)**

**Comment 1:** Although quite prominently presented in the abstract (lines 17 – 21), the change of deformation mechanism from rotation and realignment of clay-platelets dominating in the shallower section (0 – 28 mbfs) to bending and subsequent sliding / fracturing of clay particles dominating in the deeper sections is not that evident from the observations presented. Maybe a figure presenting evidence for both mechanisms next to each other would support the statement. In addition, the manuscripts describe the change of deformation mechanism observed at 28mbsf but does not elaborate on the underlying reason. Maybe the authors could discuss this aspect.

*Response: A supplementary data-18 is added to clarify both mechanisms next to each other.*
*At the shallow depth above 28mbsf, due to greater porosity, particles get enough free space for rotation to align parallel to the bedding plane. However, below 28mbsf depth, as porosity decreases, space problem arises which causes particles to deform by bending and subsequently fracturing with increase in compactional strain (Line number: 614 to 618 in revised manuscript).*

**Comment 2:** In addition to information on clay mineralogy (lines 113 – 121), additional information on bulk mineralogy should be presented, too. What is the bulk composition (not only the clay mineral assemblage)? In this context I have to admit that the information which should be transported by Supplementary data-1 is not clear to me – especially at the position where it is referred to (line 114).

*Response: Bulk composition of the analysed samples are documented in Supplementary data-4.*
*Supplementary data-1(according to old version of the manuscript) is defining the size of different clay minerals present in the samples. It is shown in the table that the mica and chlorite particles observed in the size range both <2μm (similar to clay size particles) and >2μm (similar to silt size particles). In the new version of the revised manuscript the table is represented as Supplementary data-2. To clarify this issue, we have also modified caption of the table.*

**Comment 3:** The fact that site U1480 consists of several holes, whose samples have been used for this study, should already be mentioned in line 94 instead of line 131.

*Response: DONE. (Line number: 111 in the new version of the manuscript)*

**Comment 4:** Section "sampling and methods": based on the information given, I would assume that sample preparation and analysis in Aachen and Austin was done by Ar-ion cross-section polishing succeeded by SEM investigations (i.e. BIB-SEM) with the main difference between both universities having been the instruments used (BIB polishing device: JEOL vs. LEICA; SEM: Zeiss vs. FEI). Is this correct?

*Response: Yes, this is correct.*

**Comment 5:** MAD measurements already performed should be mentioned earlier than in the "results" section.

*Response: DONE. It is also mentioned in the 'methodology' section (Line number: 209 to 218).*

**Comment 6:** Has the mineralogy presented in lines 227 – 231 been determined based on EDX element map / point analysis or is it the outcome of XRD bulk measurements (as presented in lines 113 – 121)? Qtz, Fsp, Cc, Mica und Ill are only five phases – what is the sixth phase mentioned in line 227? How was the detrital origin of the clay-size particles determined (l. 228)? Assuming that information in this paragraph are based on EDX analyses (i.e. no diffraction patterns), I wonder how it was possible to conclude that clay-size particles are dominantly illite, given that XRD analyses reveal the presence of smectite and illite/smectite.

*Response: Mineralogy of the samples are analysed using two techniques; 1) ship-board XRD analysis, and 2) EDX elemental map and point analysis on BIB cross sections. Using EDX elemental map or point analysis, it is observed that six mineral phases are dominant in the samples; these are- Quartz, feldspar (K-feldspar, Na-feldspar), calcite, pyrite, micas (muscovite, biotite, and chlorite) and clay. From ship-board XRD analysis we came to know that clay-size particles are dominantly Illite in composition. To clarify this issue, we have modified the main text (Line number: 275 to 278 (in revised version of the manuscript)).*

**Comment 7:** Given that segmentation of SE2 images only gives pores without additional information on the kind of pore, I wonder how the amount of intergranular pores (>99%) could have been determined and if this value can be considered to be representative or just a qualitative assessment (lines 241 – 242).

*Response: This is a qualitative assessment, based on the rarity of intragranular pores.*

**Comment 8**: Exchange (1) and (2) in line 254 to be consistent with Fig. 4 and lines 251/252 and 255, respectively.

*Response: DONE. The line number in the revised manuscript is 296 to 297.*

**Comment 8:** Contacts "EE", "EF" and "FF" should be highlighted in Figures 5, 7 and Supplement 13.

*Response: DONE.*

**Comment 9**: To be consistent with the information given in lines 281 – 283, a similar information should be given at some point in lines 259 – 267.

*Response: Taken care off. (Line number in revised manuscript 308)*

**Comment 10:** Unfortunately, I was not able to extract the bulk density of the sediment from the reference McNeill et al. (2017) to calculate the vertical effective stress (cf. line 373). Maybe the authors could provide these values in the manuscript / the supplement?

*Response: DONE. (Line number in revised manuscript 412 and 413).*

**Comment 11:** The statement given in lines 416/417 that "shallow depth samples are richer in smectite compared to the deeper samples" should be somewaht adjusted, because Table 1 shows that it is only true with respect to the difference between Unit I and Unit II, not with respect to Unit III.

*Response: To clarify this aspect we have modified Fig.2 where we added smectite content vs depth plot for the analysed samples.*

**Comment 12:** It appears that porosities at around 1300mbsf tend to increase – is this correct?

*Response: Yes, this is correct. Around 1300mbsf MAD porosity increases due to greater amount of smectite content in the horizon (Unit-IIIA).*

**Comment 13:** There are a bunch of comments and suggestions regarding figures and supplementary data.

*Response: DONE. These are changed according to the suggestions of both reviewers.*

---

## Referee Report (RR1)

Dear Federico Rossetti,

Please find below my review of the revised version of the manuscript "Mechanical compaction mechanisms in the input sediments of the Sumatra Subduction Complex – insights from microstructural analysis of cores from IODP Expedition 362" submitted by Sivaji Lahiri, Kitty L. Milliken, Peter Vrolijk, Guillaume Desbois, Janos L. Urai.

Most but not all of the suggested changes to the initial manuscript have been accepted by the authors. However, I still recommend to strongly considering some modifications to the initial version. In addition, some aspects, which have not been part of the initial manuscript, require some modifications according to my point of view. These points are outlined below.

Kind regards,

Bernhard Schuck

1) Referring to "comment 2" and "comment 6" in my review of the initial manuscript: I highly appreciate that the revised manuscripts is substantially more specific about the samples' mineralogy and the methods used to analyse it. However, while lines 273/274 specify that EDX analyses revealed the presence of K- and Na-feldspar, results of XRD analyses presented in table 1 only give the presence of plagioclase. This difference should be at least explicitly mentioned or discussed.
   The reference McNeil et al. (2017; https://doi.org/10.14379/iodp.proc.362.102.2017) actually only refers to the methodology applied to study the bulk rock composition. Page 10 of this methods report does not only provide values to assess the errors in XRD analysis but also states that "the method [of XRD] described is semiquantitative and results should be interpreted with caution". The authors should provide these information on errors and limitations of the performed XRD analyses in the manuscript.
   Furthermore, the supplementary material providing bulk rock compositions has a doi (https://doi.org/10.14379/iodp.proc.362supp.2017) different from the one indicated in the manuscript. Therefore, I suggest to use McNeil et al., 2017a and b as references.
   Using two digits to present "quantitative" bulk rock compositions suggests an accuracy of the results which is not justified – especially considered that the reference (McNeil et al., 2017b) only uses one digit to present bulk rock composition. For the same reasons I recommend to not provide any digits at all to present the bulk rock composition in table 1.

2) Referring to "comment 4" in my review of the initial manuscript: As far as I remember the Zeiss Supra series uses a field emission gun. There I suggest to modify lines 157 – 161 as follows to clarify the infrastructure used for microscopic analyse:

   "Samples have been prepared using Ar-ion cross-section polishing and analysed using a scanning electron microscope equipped with a field emission gun (BIB-SE technique). "The first sample set (33 mud samples; depth 1.24 to 1300 mbsf) was prepared and analysed at RWTH Aachen University, Germany. The second sample set (22 samples; depths 6.25 to 1493.30 mbsf) was prepared and studied at the Bureau of Economic Geology (BEG) at the University of Texas at Austin."

   Please note that the fact that the Zeiss is a FE-SEM implies that minor corrections to lines 327/328.

3) Referring to "comment 10" in my review of the initial manuscript: Please add the following sentence after the first sentence in line 305:

   "From this section four samples have been analysed in Aachen as well as in Austin, respectively."

4) Referring to "comment 11" in my review of the initial manuscript and to "comment 1" of this review: please modify the reference "McNeil et al. 2017" to "McNeil et al. 2017b (https://doi.org/10.14379/iodp.proc.362supp.2017) to make it easier for readers to also obtain the data used.

5) Referring to "comment 12" in my review of the initial manuscript: I still do not see how Figures 2a & b (and Table 1) support the statement that "Shallow samples from Unit-I are richer in smectite content than the deeper samples" (line 466). Both, the figures and tables show that the indeed smectite content decreases below Unit I. However, below approx. 400 mbsl smectite content increases / is elevated compared to the section between 28 and approx. 400 mbsl (see also lines 136/137 and 631/632). Therefore, I suggest to modify the statement in line 466 and discuss this observation.
   While doing so, I strongly recommend to also taking Figures 2e & f into account, which might indicate increasing porosities towards the bottom of the drillhole. Given one of the major outcomes of the study presented (cf. lines 20/21) this observation should be discussed.

6) Line 580: Please rephrase the sentence.

   Figure 1:

   - The line indicating the location of the seismic profile (b) on the satellite image (a) is yellow, not red. Please correct.
   - Please provide the orientation of the seismic profile (i.e. "SW" and "NE", respectively)

   Figure 12g:

   Still the "increase in preferred alignment of the long axes of pores" cannot be seen on the conceptual sketch, i.e. the pores' long axes appear to already have a preferred alignment in stage one, which does not change in succeeding stages. The figure should be modified to emphasize the proposed increase in preferred alignment more clearly.

---

## Author Response (AR2)

To,

Prof. Federico Rossetti,

Editor, Solid earth

*Sub: Resubmission of Revised Manuscript (SE-2022-11)*

Dear Prof. Rossetti,

Thank you for the review reports on our manuscript entitled "**Mechanical compaction mechanisms in the input sediments of the Sumatra Subduction Complex- insights from microstructural analysis of cores from IODP Expedition- 362**" written by Lahiri, S.; Milliken, K. L.; Vrolijk, P., Desbois, G.; and Urai, J. L. "Minor Revision" was recommended based on the review report. **Reviewer#1** (Dr. Dave Dewhurst) stated regarding the manuscript that, 'this manuscript is vastly improved and pretty much ready to go with just the clay mineralogy clarification to go along with some minor cosmetic tidying up. Really good work and great images'. **Reviewer#2** (Dr. Bernhard Schuck) commented that "the manuscript is well-written and address a topic relevant to solid earth". However, the reviewers raised a few questions regarding clay mineralogy, bulk rock mineralogy and also suggested some minor changes.

We appreciate the constructive review by the reviewers, and in the revised manuscript we have addressed all the issues raised by them. *In the revised manuscript, all the changes based on suggestions of the reviewers are highlighted in green color (reviewer#1, Dave Dewhurst) and yellow color (reviewer#2, Bernhard Schuck).*

I hope that the revised version will be accepted for publication in the *Solid Earth*.

Thanking you,

Yours sincerely

Sivaji Lahiri

(For the authors)

**Comments by Dave Dewhurst**

**Comment 1:** My main concern with the manuscript last time was that the description of the clay mineralogy of the sediments was somewhat confusing and this unfortunately is still the case. As one of the senior authors (Milliken) is currently visiting my office in CSIRO in Perth, I have discussed this issue with her, and it seems that this can be clarified. I will still write my comment here however so that the authors can address it in the final version.

The text from lines 131-144 states initially that smectite, illite, kaolinite and chlorite are present in the samples. However, it then goes on to discuss mixed layer illite-smectite, which doesn't appear in table 1 at all. I know that the XRD data is not the crux of this manuscript and comes from the Rosenberger paper, so I went and checked that too, but it isn't completely clear in that paper either. So, what I am left wondering is whether the "smectite" quoted in table 1 is pure smectite or whether it's mixed layer illite-smectite (or both)?

From the Rosenberger paper looking at the diffractograms presented there and how things are worded, I would guess that the "smectite" quoted is all mixed layer illite-smectite and that there isn't any pure smectite in these sediments. Or if that is incorrect, is all the I-S hiding under the "illite" number given in table 1, or is that just illite and all the various micas? I have seen a few shales which do have pure smectite, mixed layer I-S and illite and these are routinely specified as separate phases when they all occur together in the XRD work we do here. Hence, I have ended up a bit confused. I gather that the XRD was done in two batches, one shipboard for bulk and one onshore for detailed clay mineralogy. Perhaps a bit of clarity around the different techniques used and different timings would be helpful so the reader can clear understand what has been done and how the quantitative determinations were made.

The final thing I am a bit confused on (I'm having one of those days I think!! ☺), is on lines 137-139, where it is stated that "There is a weak increase in the expandability of I-S with depth" but "no change in the amount of illite in mixed layer I-S". Presumably the increase in expandability with depth results from increasing smectite in I-S with depth (which I gather is from increasing volcanogenic components deeper in the sequence, might be useful to note in the discussion maybe?), so if smectite is increasing, shouldn't illite be decreasing? Sorry if I am missing something obvious.

*Response:* Further explanation of the data reported in Rosenberger et al. is added in Lines 136-147. Basically, the amount of illite/smectite in samples, which are smectite-rich (highly expandable), are first defined in terms of the smectite used in the reference samples for quantification. A separate analytical step on glycol-saturated samples was performed to determine the amount of illite in mixed-layer I/S (defined in terms of Expandability). As now stated, the accumulation of clay mineral data and how they are combined into a comprehensive understanding should now be clear (line number 132 to 156 in the new version of the manuscript).

**Comment 2:** Lines 64-66 and 70-74, the previous investigations quoted have been divided up into two groups, "lab experiments" and "natural samples". However, some of the lab experiment references quoted were actually performed on natural samples, which makes things a bit unclear. Perhaps this could be better phrased as (1) experimental compaction in the laboratory and (2) in situ compaction with depth, or some such. Need to rephrase at lines 70-74 as not all the lab tests were done on artificial clay packs.

*Response: DONE.* Line number 65 to 73 in the new version of the manuscript.

**Comment 3:** L117-118, Units I to VI are not all visible in Figure 1c as the figure seems to be chopped off at the bottom only getting to unit IV.

*Response: DONE.* We have modified Fig.1c.

**Comment 4:** L230 and other places through the manuscript, the magnification is written out in two different ways, in places for example with x 10,000 while in other spots 10kx. I would stick with writing things out at x 10,000 for consistency. I have noted a few spots in the marked-up version that should be made consistent.

*Response: DONE.* Line number 244, 245 to 284 in the new version of the manuscript.

**Comment 5:** L285, the authors are discussing figure 3b and one observation that should be noted is that although there is a correlation between the two porosity measurements, there is a large difference in the absolute values and that this difference increases with increasing porosity. This observation becomes useful to support interpretations later in the discussion. I have written this comment at L215 on the marked manuscript but decided it would be a better fit at L285.

*Response: DONE.* We have added the line. Line number 302 to 304 in the new version of the manuscript.

**Comment 6:** L320-325, I think the figure numbers are incorrect in this paragraph.

*Response: DONE.* **Corrected.** Line numbers 341-345 in the new version of the manuscript.

**Comment 7:** L337, this supplementary data figure number also appears to be incorrect.

*Response: DONE.* Line number 359 in the new version of the manuscript.

**Comment 8:** L430-432, in regard to the previous studies noting mismatches between MAD and BIB-SEM, what did they actually conclude were the causes?

*Response: DONE.* Line numbers 460-463 in the new version of the manuscript.

**Comment 9:** L442, the correlation coefficients quoted don't seem to match what is on the figure.

*Response: DONE.* Line number 473 in the new version of the manuscript.

**Comment 10:** L469, A comment could be added here stemming from point 5 above that the results are also consistent with the difference between the porosity measurements getting larger as porosity increases.

*Response: DONE.* Line number 501-502 in the new version of the manuscript.

**Comment 11:** There are a few typos and grammatical suggestions which are on the accompanying marked manuscript. For the main text, the references are correct with one exception at the bottom of page 3 (Hippchen and Hyndman is not in the list). However, there are a number of references in the supplementary text which are not in the list and these are on the marked manuscript.

*Response: DONE.*

**Comment 12:** On figure 3a, the 1:1 line is not in the correct place. Both porosity axes also need the units displayed.

*Response: DONE.*

**Comment 13:** Figures 12a and 12b also need the porosity units on those axes. What do the different coloured symbols represent in Figure 12a?

*Response: DONE.* Green, orange and blue colored symbols represent samples from Unit-I, II and III respectively. To clarify this issue, we have described the color codes in the Fig.12a and also mentioned in caption.

**Comment 14:** Perhaps the legend to table 1 could note what Pc/pc+g is as a column title. Perhaps also add in that bulk measurements were shipboard, and that the clay mineralogy was done on the < 2 micron fraction onshore on a different instrument.

*Response: DONE.* A small description is added in the caption of the table.

**Comment 15:** In Figure S1, the very ends of the axes need reversing for kaolinite/chlorite and illite content, otherwise they are reading as higher than total clay content.

*Response: DONE.*

**Comment 16:** The supplementary written documents have a lot of typos and some referencing issues. These are on the marked manuscript.

*Response: DONE.*

**Comments by Bernhard Schuck**

**Comment 1:** Referring to "comment 2" and "comment 6" in my review of the initial manuscript: I highly appreciate that the revised manuscripts is substantially more specific about the samples' mineralogy and the methods used to analyse it. However, while lines 273/274 specify that EDX analyses revealed the presence of K- and Na-feldspar, results of XRD analyses presented in table 1 only give the presence of plagioclase. This difference should be at least explicitly mentioned or discussed.

*Response: DONE.* Line numbers 291 to 294 in new manuscript.

**Comment 2:** The reference McNeil et al. (2017; https://doi.org/10.14379/iodp.proc. 362. 102.2017) actually only refers to the methodology applied to study the bulk rock composition. Page 10 of this methods report does not only provide values to assess the errors in XRD analysis but also states that "the method [of XRD] described is semiquantitative and results should be interpreted with caution". The authors should provide these information on errors and limitations of the performed XRD analyses in the manuscript.

*Response: DONE.* Line numbers 156-158 in the new version of the manuscript.

**Comment 3:** Furthermore, the supplementary material providing bulk rock compositions has a doi (https://doi.org/10.14379/iodp.proc.362supp.2017) different from the one indicated in the manuscript. Therefore, I suggest to use McNeil et al., 2017a and b as references.
*Response: DONE.*

**Comment 4:** Using two digits to present "quantitative" bulk rock compositions suggests an accuracy of the results which is not justified – especially considered that the reference (McNeil et al., 2017b) only uses one digit to present bulk rock composition. For the same reasons I recommend to not provide any digits at all to present the bulk rock composition in table 1.

*Response: DONE.*

**Comment 5:** Referring to "comment 4" in my review of the initial manuscript: As far as I remember the Zeiss Supra series uses a field emission gun. There I suggest to modify lines 157 – 161 as follows to clarify the infrastructure used for microscopic analyse:

"Samples have been prepared using Ar-ion cross-section polishing and analysed using a scanning electron microscope equipped with a field emission gun (BIB-SE technique). "The first sample set (33 mud samples; depth 1.24 to 1300 mbsf) was prepared and analysed at RWTH Aachen University, Germany. The second sample set (22 samples; depths 6.25 to 1493.30 mbsf) was prepared and studied at the Bureau of Economic Geology (BEG) at the University of Texas at Austin."

*Response: DONE.* Line number 171-175 in the revised manuscript.

Please note that the fact that the Zeiss is a FE-SEM implies that minor corrections to lines 327/328.

*Response: DONE.* Line number 350 in the revised manuscript.

**Comment 6:** Referring to "comment 10" in my review of the initial manuscript: Please add the following sentence after the first sentence in line 305:
"From this section four samples have been analysed in Aachen as well as in Austin, respectively."

*Response: DONE.* Line 324-326 in new version of the manuscript.

**Comment 7:** Referring to "comment 11" in my review of the initial manuscript and to "comment 1" of this review: please modify the reference "McNeil et al. 2017" to "McNeil et al. 2017b (https://doi.org/10.14379/iodp.proc.362supp.2017) to make it easier for readers to also obtain the data used.

*Response: DONE.*

**Comment 8:** Referring to "comment 12" in my review of the initial manuscript: I still do not see how Figures 2a & b (and Table 1) support the statement that "Shallow samples from Unit-I are richer in smectite content than the deeper samples" (line 466). Both, the figures and tables show that the indeed smectite content decreases below Unit I. However, below approx. 400 mbsl smectite content increases / is elevated compared to the section between 28 and approx. 400 mbsl (see also lines 136/137 and 631/632). Therefore, I suggest to modify the statement in line 466 and discuss this observation.

*Response: DONE.* line 497-498 in the new version of the manuscript.

While doing so, I strongly recommend to also taking Figures 2e & f into account, which might indicate increasing porosities towards the bottom of the drillhole. Given one of the major outcomes of the study presented (cf. lines 20/21) this observation should be discussed.
*Response: DONE.* line 373-378 in the new version of the manuscript.

**Comment 9:** Line 580: Please rephrase the sentence.

*Response: DONE.* line 613-615 in the new version of the manuscript.

**Comment 10:**
Figure 1:
- The line indicating the location of the seismic profile (b) on the satellite image (a) is yellow, not red. Please correct.
- Please provide the orientation of the seismic profile (i.e. "SW" and "NE", respectively)
*Response: DONE.*

Figure 12g:
Still the "increase in preferred alignment of the long axes of pores" cannot be seen on the conceptual sketch, i.e. the pores' long axes appear to already have a preferred alignment in stage one, which does not change in succeeding stages. The figure should be modified to emphasize the proposed increase in preferred alignment more clearly.

*Response: DONE.* We have redrawn the Figure.12g. (Fig.14g in new manuscript).